# State-of-the-Art Mobile Radiation Detection Systems for Different Scenarios

**DOI:** 10.3390/s21041051

**Published:** 2021-02-04

**Authors:** Luís Marques, Alberto Vale, Pedro Vaz

**Affiliations:** 1Centro de Investigação da Academia da Força Aérea, Academia da Força Aérea, Instituto Universitário Militar, Granja do Marquês, 2715-021 Pêro Pinheiro, Portugal; 2Instituto de Plasmas e Fusão Nuclear, Instituto Superior Técnico, Universidade de Lisboa, Av. Rovisco Pais 1, 1049-001 Lisboa, Portugal; avale@ipfn.tecnico.ulisboa.pt; 3Centro de Ciências e Tecnologias Nucleares, Instituto Superior Técnico, Universidade de Lisboa, Estrada Nacional 10 (km 139.7), 2695-066 Bobadela, Portugal; pedrovaz@ctn.tecnico.ulisboa.pt

**Keywords:** mobile radiation detection systems, NORM, radiological emergencies, nuclear accidents, illicit trafficking, accelerators, targets and irradiation facilities, gamma and neutron detectors

## Abstract

In the last decade, the development of more compact and lightweight radiation detection systems led to their application in handheld and small unmanned systems, particularly air-based platforms. Examples of improvements are: the use of silicon photomultiplier-based scintillators, new scintillating crystals, compact dual-mode detectors (gamma/neutron), data fusion, mobile sensor networks, cooperative detection and search. Gamma cameras and dual-particle cameras are increasingly being used for source location. This study reviews and discusses the research advancements in the field of gamma-ray and neutron measurements using mobile radiation detection systems since the Fukushima nuclear accident. Four scenarios are considered: radiological and nuclear accidents and emergencies; illicit traffic of special nuclear materials and radioactive materials; nuclear, accelerator, targets, and irradiation facilities; and naturally occurring radioactive materials monitoring-related activities. The work presented in this paper aims to: compile and review information on the radiation detection systems, contextual sensors and platforms used for each scenario; assess their advantages and limitations, looking prospectively to new research and challenges in the field; and support the decision making of national radioprotection agencies and response teams in respect to adequate detection system for each scenario. For that, an extensive literature review was conducted.

## 1. Introduction

Radioactive materials, radioactive sources, and radiation sources are ubiquitous; they are used in practically all sectors, made of different radionuclides, emit different types of ionizing radiation (gamma rays, alpha and beta particles, neutrons), and are characterized by their activity (number of disintegrations per second). Special nuclear materials (SNM) are present in different civilian and military facilities, namely along the nuclear fuel cycle (nuclear fission reactors for electricity production, nuclear fuel fabrication, re-processing and storage facilities, etc.) and in nuclear propulsion vessels (namely submarines and carriers).

Concerns about potential malevolent acts involving illicit trafficking of radioactive materials and SNM have increased with the heightened awareness of international terrorism where a variety of radiological and nuclear (RN) disaster scenarios could occur including threats in urban areas. An example of a non-precedent terrorist act which showed the terrorists’ ability and willingness to use any means to achieve their goals, was the attack to the World Trade Center towers (11 September 2001). This event changed the paradigm of security and defense worldwide. Another example, now related to the malevolent use of radioactive material, was the poisoning of Alexander Litvinenko (2006) with polonium-210 [1].

According to the International Atomic Energy Agency (IAEA) [2], between 1993 and 2019, a total of 3686 incidents, of which 290 involved a confirmed or likely act of trafficking or malicious use, are reported in the illegal traffic database. Other incidents reported in the IAEA database related to safety and security issues associated to radioactive materials involve, inter alia, theft or loss of radioactive sources, unauthorized disposal (sources entering the scrap metal industry), unauthorized shipment of contaminated scrap metal, and the discovery of radioactive sources (orphan sources, out of regulatory control).

Radiation portal monitors (RPM) are often used for the prevention of illicit traffic of SNM and radioactive materials, as well as the inadvertent movement of radioactive material. They are located at appropriate checkpoints (e.g., border crossings, airports and seaports) to measure gamma and neutron radiation. Other radiation detection equipment are personal radiation detectors (PRDs) used by front line officers (FLO), hand-held gamma and neutron search detectors, and hand-held radionuclide identification devices. Hand-held instruments can be used as primary detection systems (greater flexibility), or as a secondary search device, for example, to validate a reading from an RPM or PRD [3].

Due to the many potential entryways or transportation modes in a country [4] and inherent difficulty to detect such materials, particularly SNM (normally weak sources, possibly shielded or masked) [5], it is necessary to improve the detection probability to reduce the wrongdoer’s success. The use of portable radiation portal monitors (PRPM), helped to mitigate the lack of devices in other strategic points (depending on the potential threats). Since the a priori location of the PRPMs is unknown, it has also a deterrence effect on potential terrorist acts. However, the "portability" of these devices only means that they can be disassembled into a case for the subsequent transport. The PRPM are normally made with plastic scintillation detectors and can also be used for RN accidents and emergencies (people and vehicles monitoring) [6]. Considering other detection strategies, Cazalas [7] suggested an RN threat detection solution based on a network of radiation detectors at existing road traffic-monitoring system locations (e.g., stop light or red light cameras), while Coogan et al. [4] concluded that mobile radiation detection (MRD) technology with at least one-tenth of the RPM efficiency can have the same or greater impact on the wrongdoer’s success rate. MRDs have the advantage of performing discrete operations and can be deployed to major thoroughfares or to protect a potential target. However, there is a lower control over the geometry, and the MRD performance is highly dependent on their placement.

In the same way, to monitor an unintended release of radionuclides, as is the case of a nuclear reactor accident, there are fixed online radiation monitoring systems in the vicinity of the power plant. However, the information gathered by these devices are confined to their location and, in the situation of a natural disaster, like happened in Fukushima, these detectors may fail [8]. This can lead to a lack of information about the radiation levels and consequently in bad, inaccurate, and delayed decisions.

The rapid deployment of an MRD system can help to monitor locations not covered by the fixed network or be the main detection system in an RN event [9], for example, to evaluate the activity and direction of the radioactive plume spread on the ground (due to fallout) or in the air (e.g., using an air sampler). This mobility allows covererage of large areas and reduction of survey times.

Other scenarios where it is important the use of mobile detection systems are: (i) areas with high concentration of naturally occurring radioactive materials (NORMs), normally associated with large survey areas and dose rates near background radiation, and (ii) the inspection, maintenance and repair activities in nuclear energy facilities, particle accelerators, targets, and irradiation facilities, characterized by dose rates ranging from low to high, and, eventually, the presence of high magnetic fields and high temperatures.

The use of mobile detection systems may allow to improve the detection efficiency of RN materials by decreasing the distance between detector and source and the effects of air or materials’ attenuation.

For each scenario described, one might be interested in the location, identification, and quantification of a radioactive source or the mapping of a contaminated area. Due to their long range in air, gamma-rays and neutrons are preferable indicated for mobile detection systems.

This review article encompasses mobile detection systems carried by a person (e.g., handheld and backpack equipment), as well as detectors coupled to ground-based vehicles (e.g., trucks, vans, and cars) and air-based vehicles, like fixed-wing or rotatory wing aircrafts. Both ground-based and air-based vehicles can be manned or unmanned.

Since unmanned vehicles can be used in dull, dirty and dangerous missions, their use is of great interest in RN events, particularly when the radiation field is unknown (e.g., RN accident or incident and RN threat) or poses radiological risk to humans.

Despite the use of unmanned ground vehicles (UGVs) in nuclear accidents as in Chernobyl (1986) and Fukushima (2011) [10], only after the Fukushima accident did unmanned aerial vehicles (UAVs) start to be used as platforms for radiation monitoring and mapping [11].

During the last decade, emerging radiation detection technologies allowed smaller and cheaper radiation sensors, as is the case of: novel gamma-ray scintillating crystals with increasing efficiency and better energy resolution (e.g., standard and enhanced lanthanum bromide), novel neutron detectors with high efficiency and good gamma-ray discrimination, sensors sensitive to either neutron and gamma radiation (dual-mode sensors), the use of compact semiconductor photosensors instead of the fragile and heavier photomultipliers (PMTs), compact and low power data acquisition systems, smart detector instruments that allow the data fusion of multiple radiological and non-radiological sensors (contextual sensors) [12], portable and lightweight gamma cameras, and the new dual particle cameras (gamma and neutrons). Additionally noteworthy is the growing demand for low weight, low power consumption and high radiation tolerance detectors in the aerospace industry, particularly in space technology where some detectors were already deployed [13,14].

The recent developments in robotics allowed the integration of such compact radiation detection systems in small unmanned systems. The use of such technology with the help of new algorithms resulted in improvements in the reliability of source detection, location and identification reducing in the same way the false alarm rates. An important new feature is the autonomous localization of a radiation source [12].

A new era started with the first use of small unmanned aircraft systems (SUAS) in a scenario following a nuclear accident (Fukushima, 2011). Despite the new challenges of flying at low altitude, such as in urban environments, the dose risks to humans were eliminated, and there was a significant improvement in the spatial resolution of the radiation mapping compared to manned aircrafts. Since then, new technologies appeared, involving the use of low-cost UAVs (e.g., swarm UAVs) for source localization and mapping or the cooperative navigation between different unmanned platforms.

Unlike in a laboratory, MRD systems measurements are performed in a non-controlled environment. For example, indoor environments are characterized by the possible global navigation satellite system (GNSS) signal denial and obstacles (e.g., stairs, doors and narrow passages), while outdoor environments are characterized by the weather influence (e.g., rain, wind, and atmospheric pressure) and obstacles, like tall vegetation, sea lines, steep slopes, and artificial constructions. A special challenging outdoor environment is an urban area, which may also cause GNSS signal denial (e.g., between tall buildings) [15].

While there are some papers already published about these topics, they cover only a part of the scope of this work, thus missing the interconnection between different scenarios or the reference of either neutron or gamma detection systems. Kumar et al. (2020) [16] presented the recent developments in radiation detection systems used in ground and air-based platforms for emergency radiation monitoring scenarios (radiation contamination resulting from nuclear accidents); Connor et al. [17] described the aerial platforms used in airborne radiation mapping and perspectives; Schneider et al. (2015) [18] presented the unmanned systems with potential to be used for radiation measurements and sampling; Ihantola et al. [12,19] describes the recent detection technologies for nuclear security and their impact; Cieślak et al. [20] and Hamrashdi et al. [21] presented a review of gamma and neutron imagers, the latter also included a review of passive gamma ray detection.

This paper aimed at describing the salient developments in mobile radiation detection systems coupled to ground-based (handheld equipment included) and air-based platforms from the era after the Fukushima Daiichi nuclear power plant accident (FDNPP), considering four reference scenarios. The advantages and limitations of each detection system are also analyzed, highlighting the challenges and future research needed in these fields.

The literature review used three different bibliographic databases, Scopus, Web of Science, and Google Scholar, and it was limited to articles published between the era post-Fukushima accident and the end of July 2020. Different keywords were used, like mobile, radiation, gamma imaging, and nuclear, and the subject area was limited to engineering, physics, computer science, and environmental science. For the search, original peer-reviewed research articles and literature reviews were included. In total, more than 200 bibliographic references were identified and analyzed. Data were compiled from books, published journal articles, conference proceedings, and grey literature, in particular: technical reports from international agencies, like the IAEA, or national agencies, such as the United States Department of Homeland Security, and information obtained from manufacturer websites (e.g., Saint-Gobain) that are not normally subjected to peer review and, therefore, may contain biased data. However, all the data were cross-checked to guarantee validity of the conclusions.

The remainder of this paper is divided into six sections. The “Scenarios” section briefly describes the four scenarios considered in this article. The “Mobile platforms” section gives an insight of the different platforms that can be used for radiation detection, as well as challenges and ongoing research. The “Mobile Radiation Detection Systems” section covers the existing gamma and neutron detectors described in the literature. In the “Results and Discussion” section, the benefits and limitations of the different combinations of mobile platform, detection sensors, and contextual sensors, considering the four scenarios, are analyzed. In the last section, conclusions are presented, and prospective views are provided.

## 2. Scenarios

The choice of a certain mobile radiation detection system depends on the characteristics of the scenario. In this article, four scenario types will be considered and analyzed, which are detailed in the next subsections, namely:RN accidents and emergencies (hereafter designated scenario A).Illicit trafficking of SNM and radioactive materials (hereafter designated scenario B).Nuclear, accelerator, targets, and irradiation facilities (hereafter designated scenario C).Detection, monitoring, and identification of NORM (hereafter designated scenario D).

### 2.1. Radiological and Nuclear Accidents and Emergencies—Scenario A

This scenario is related to the response to an intentional release of radioactive material (e.g., radiological threat) or a non-intentional release of radioactive material, like the major nuclear accident.

During the Fukushima nuclear accident, radioactive isotopes (mainly ^137^Cs, ^134^Cs, and ^131^I) were released to the atmosphere with formation of a radioactive plume that, afterwards, traveled and later deposited in the surface. Concentrations of these radionuclides still remain in the soils. Therefore, it soon became important to identify radiation hotspots and measure the effectiveness of the decontamination operations (remediation process) [22].

Another example of an accidental release of radioactive materials was the environment release of ^106^Ru in September 2017 that has been detected all over Europe [23].

Since the spread of radioactive material can easily extend into several tens of kilometers, in the event of a nuclear catastrophe, a ground-based monitoring system may not be practical nor feasible due to financial constraints and possible safety- and security-related issues. Due to the non-availability of proper roads in remote locations, thick vegetation, abrupt slopes, and water passages, the ground-based may not be possible or should be confined to small regions. An alternative is airborne detection systems using real-time monitoring methods at safe altitudes to monitor high levels of radiation [17].

An important publication was released in 1999 by IAEA [24] relative to generic procedures for radiological monitoring in a nuclear or radiological emergency, to help Member States in developing appropriate radiation monitoring programs, procedures, and standards, as well as providing practical guidance for environmental and source monitoring, during a nuclear or other radiological emergency.

Post-accident decommissioning of a nuclear facility is also an important issue since it presents many differences compared to normal decommissioning. For example, in an accident scenario, the radiological and physical characterization is normally compromised by limitations on access (e.g., physical disruption of normal access routes) in regions where inspections, measurements, or sampling are necessary. Moreover, the presence of high radiation fields may limit the human access, making it necessary to use special remote tooling. After a nuclear accident with catastrophic fuel failure, the main contributor for the gamma radiation field will be ^137^Cs ( 30 year half-life), leading to a slower radioactive decay compared to a normal reactor shut down, which is dominated by ^60^Co ( 5.3 year half-life) in the near term. Another characteristic of a post-accident scenario is the contamination of the power plant surroundings with actinides (radioactive elements with atomic numbers 89 to 103) due to fuel rupture [25].

In this scenario, one might be interested in detecting, localizing, quantifying, and identifying the released source(s) (hotspots or contaminated areas), or just obtaining the map of the radionuclides distribution. In order to monitor the distribution of radionuclides present in a given contaminated area (e.g., understand the effectiveness of a remediation processes or the mobility of radionuclides in soils), a mapping over time to compare changes in their concentrations (e.g., dose rates) can be performed.

### 2.2. Illicit Trafficking of SNM and Radioactive Materials—Scenario B

This scenario is focused on the prevention of malicious use of enriched nuclear material into improvised nuclear devices or the use of radioactive materials (sources) to produce radiological dispersal devices (RDD) (aka “dirty bombs”). Despite the fact that the probability of an RDD attack its unknowable, they have a tremendous impact on safety, economic, and psychological effects, being sometimes referred to as weapons of mass disruption [26].

Nuclear materials (mainly plutonium and uranium isotopes) can be obtained from countries that hold nuclear weapons, have nuclear weapons programs, or operate internal enrichment or reprocessing facilities [7].

On the other side, radioactive materials may be obtained from radioactive and radiation sources used in industry (e.g., in oil well logging, irradiators for sterilization of medical devices and food products, or in thermoelectric generators) and medicine (e.g., in blood irradiators and brachytherapy seeds in hospitals). Some relevant radioisotopes are the gamma-emitters ^60^Co, ^137^Cs, and ^192^Ir. An example of a pure beta emitter is ^90^Sr. Either emitters can be used to attack governmental or financial centers, population, or critical infrastructure.

To prevent the illicit trafficking of radioactive and SNM or the inadvertent movement of radioactive material, normally, there are radiation detection systems (e.g., RPMs and/or mobile radiation monitors) deployed on land, sea gantries, and airports. This is part of the country’s strategy for homeland security, which involves not only fighting the smuggling of nuclear and radioactive material but also other dangerous substances, such as biological and chemical agents, or explosives.

Detection of SNM typically relies on gamma and neutron radiation. The radiation signals detected from these materials are relatively weak and specially difficult to detect at distance (e.g., plutonium and highly enriched uranium) [7].

Detection and interpretation of gamma-ray signals are typically easier than that for neutrons, due to the detection equipment used, relative ease of obtaining gamma spectrometry data, the need for radiation-type discrimination in neutron detection, and use of thermalization mediums for neutron moderation and detection. However, gamma-ray detection of threat materials is complicated due to the natural background, approximately an order of magnitude higher than for neutrons [7], originated by NORM radionuclides (e.g., transportation of bananas or cat litter—^40^K and ^232^Th decay chain, respectively). NORM may cause unacceptable rates of false alarms in detection systems or may be utilized to mask the signal from threat materials. Additional complications are due to the relative ease of gamma-ray shielding, which may be accomplished with a few centimeters of high-Z materials, such as lead, or even by the structure of the vehicles.

The detection instruments may be divided into two groups: (i) passive and (ii) active. Since passive detection systems are based on the direct measurement of the natural emissions of radiation, in some cases, they are not enough to detect concealed SNM, for example, weak sources, like highly enriched uranium or possible shielded sources. Thus, the use of active detection systems should be considered. The principle of operation of the active detection equipment is based on impinging the suspect object with radiation, like X-rays or gamma-rays (radiography) or neutrons (active interrogation). Radiography allows distinguishing between low and high density materials (shielding detection), while the active interrogation can detect nuclear materials (shielded or not shielded) by measuring their radiation signature in response to incident neutrons [27]. Another way to discover the presence of dense materials in transit cargo is by using tomographic imaging with cosmic ray muons [28].

In scenario B, the main goal of MRD systems is to detect, quantify, and localize SNM sources and other radioactive material resultant of illicit tracking or inadvertent movement. Since these sources are normally weak (e.g., SNM) and the detector-source distance can be high (up to 100 m), normally, high efficiency detection systems are necessary. To distinguish threat sources from NORM or other medical isotopes, identification of the source is also needed. In the next subsection, the existent standards for the detection systems used in security applications are described.

When new detection systems are developed, they should first be analyzed accordingly to existing standards, specifying performance requirements and test methods. On the other hand, when novel technology has a significant impact on the instruments performance or introduces new features, it is necessary to revisit the existing standards, e.g., the first use of scintillator material for neutron detection led to the revision of standards [19].

The International Electrotechnical Commission (IEC) and the American National Standards Institute (ANSI) have published some important standards for mobile radiation detection equipment, listed in Table 1, which describes not only radiological requirements and test methods, but also requirements, like mechanical, electrical, and magnetic related properties.

Both IEC63121:2020 [29] and ANSI N42.43-2016 [30] standards are only applied to vehicle-mounted mobile systems which travels predominantly on public roads (e.g., car, van or trailers). Therefore, these documents do not apply to detection systems mounted in other types of vehicles, like air-based platforms (e.g., planes and helicopters), trains, or boats.

In order to keep the same data formats among the detection systems, there are important standards that must be considered: IEC 62755:2012 [31] and ANSI N42.42-2012 [32], for radiation instruments used in the detection of illicit trafficking of radioactive materials; and IEC 63047:2018 [33], which defines the data format for list-mode digital data acquisition used in radiation detection and measurement. The latter allows the collation of data from one or more detectors and one or more data acquisition devices (e.g., time-correlated data) [19].

### 2.3. Nuclear, Accelerator, Targets, and Irradiation Facilities—Scenario C

This scenario addresses facilities and installations, such as nuclear energy facilities (fission and fusion), as well as high energy and high beam intensity particle accelerators, targets, and irradiation facilities, for emerging and innovative applications of nuclear technologies. Some examples are: spallation neutron sources, accelerator driven systems (ADS) for transmuting radioactive waste, and multi-MegaWatt targets, for fundamental and applied science applications.

To ensure workers and environmental safety, these nuclear technologies need to undergo periodic or unexpected inspections, maintenance, and repair. However, due to the demanding environment characterized by high dose rates (combination of proton, neutron and photons), high magnetic fields (e.g., accelerators), and high temperatures (e.g., fission reactors, spallation targets, ADS, multi-MegaWatt targets), these tasks presents a challenge to the radiation monitoring systems.

Normal decommissioning of nuclear power plants (NPP) and accelerators can also be included in this scenario. In the case of an NPP, this includes the decontamination of the facility (reduce residual radioactivity), dismantling the structures and removing contaminated materials to appropriate disposal facilities (nuclear fuel storage). In nuclear reactors, approximately 99% of the residual radioactivity is related with the fuel (removed after shutting down). The remaining is due to activation products in steel (reactor pressure vessel exposed to neutron irradiation) producing highly radioactivity iron-55, iron-59, and zinc-65 (half-lives of 2.7 years, 45 days, and 245 days, respectively); thus, 50 years after shut down, their radioactivity is very low, and the radiological risk is significantly reduced to workers. Since the decommissioning of an NPP can take up to 60 years and more than 180 commercial, experimental, or prototype reactors, as well as more than 500 research reactors and several fuel cycle facilities, have been shut down worldwide, this issue is gaining prominence [34]. In Reference [35], some R&D activities needed for the decommissioning of nuclear facilities are discussed. The optimization of robotics is suggested in areas, like 3D integrated gamma-ray and vision systems, and in the developing of flexible robots (possible mounting of different tools).

Today, particle accelerators are used in many applications as: radioisotope production, medical applications, sterilization of medical devices and food products, mineral prospecting and oil well logging, material science and applications, fundamental and applied physics research, archaeological dating, and cargo inspection. Particle accelerators facilities produce and make available beams of particles, like electrons and protons, as well as deuterons, helium nucleus, and other heavier ions. Radioactivity can be induced by primary beam interactions or by indirect interactions of secondary particles in the surrounding structural, and the specific activity of the radioactive materials produced varies significantly according to the accelerator type and setup. An example of a byproduct of accelerator-based experiments is tritium. Therefore, it is important to characterize the radiological risk, not only during normal operation but also due to beam losses, after beam shutdown, etc., as well as in the decommissioning process. Computer programs (e.g., by using Monte Carlo, deterministic or hybrid techniques) can be used to estimate radiation field intensities and accelerator structure’s activation [36].

Decommissioning of particle accelerators can be challenging since activation distribution varies significantly in a facility with the possible presence of hot spots. Even knowing that, in some accelerators, the residual activation is low, this may add difficulty in the measurements due to the very restrictive radiation dose limits to humans. Despite the fact that fusion machines are not considered as accelerators, their decommissioning processes have some similarities [36]. Moreover, techniques used in the decommissioning of fission reactors were adapted to the decontamination and decommissioning of the Tokamak Fusion Test Reactor [37].

In this scenario, the mobile detection systems can be used to detect and localize possible leaks of radioactive material and quantification of the radiation field intensity (e.g., generated by the facility operation or activation products).

### 2.4. Detection, Monitoring, and Identification of NORM—Scenario D

Since natural radiation is a major contributor to the exposure of the population to ionizing radiation, it is important to assess and understand its impact on the general public and environment radiation safety [38] and radiological risks.

The radionuclides that contribute to natural radiation can be divided into: (i) terrestrial NORM (the vast majority), isotopes that belong to the Uranium and Thorium decay series (mainly because of radon and progenies), as well as ^40^K, and (ii) Cosmogenic NORM, resultant products of the interaction of cosmic rays with the atmospheric gases (very little contribution to the dose).

Despite issues related to radon exposure in homes, particularly those built on granitic ground, the main concern is about NORM that has been altered in the course of exploitation of natural resources to meet human needs (namely construction materials and industrially processed materials).

Human activities that exploit these resources, essentially products, by-products, and/or wastes of industrial activities, may lead to enhanced concentrations of radionuclides normally referred to as technologically-enhanced NORM. For simplicity, the term NORM will be used throughout the article [39]. For example, the processing of minerals, like uranium ores, monazite (a source of rare earth minerals), and phosphate rock used to produce phosphate fertilizer, has the potential to result in radiation doses above natural background [40]. Since NORM have many pathways for reaching the humans (e.g., ingestion along the food chain, inhalation of radon isotopes and their progeny, or ingestion of airborne radioactive dust), these enhanced concentrations of NORM may present serious radiological risks. In order to avoid possible health impacts due to radiation hazards, and keep dose limits below the recommended values, necessary long-term monitoring of the radiation field intensity (dose) and identification of the sources (NORM or other) on these sites [41] is of paramount importance.

Due to the long half-lives of NORM (e.g., ^238^U—4.5 billion years, ^232^Th—14 billion years, and ^40^K—1.3 billion years), small concentrations practically exist in practically all materials. Therefore, depending on the concentration of NORM in some cargo material, it may trigger a false alarm in a portal monitor (e.g., RPM) at a border crossing. In 2003, half the nuisance alarms detected were caused by medical sources (16%) and kitty litter cargo—NORM (34%). Other cargo materials that may present significant concentrations of NORM are: Abrasives, refractory material, and raw materials, such as mined products, Brazil nuts, and bananas [42]. Since these innocent alarms may have a significant impact in the people and cargo traffic, it is necessary to improve the detection, monitoring, and identification of NORM relative to other man-made sources.

The MRD systems can be used to detect, localize, quantify, and identify NORMs. Since NORM distribution may vary from place to place, due to the variation of mineral content in soils or due to human activities (e.g., ore extraction), it is generally also necessary to map an area and monitor the radiation field intensity and NORM concentrations over time.

## 3. Mobile Platforms

When choosing a mobile platform to carry a radiation detection system for a specific task or scenario, it is important to know what are the requirements that must be taken into account. Requirements, such as the weather sensitivity of the vehicle, payload capacity, and how that affects its performance (e.g., range, operational time), cost (initial investment and operational cost), ease of operation, ease of decontamination, and spatial resolution attainable (e.g., for mapping purposes) [18], can impact on the quality and effectiveness of the radiation measurements. Therefore, to choose the right mobile platform, it is necessary to know its advantages and limitations.

Mobile platforms can be divided into ground-based and air-based. Each platform may be either manned or unmanned.

Compared to manned ground vehicles or aircrafts, unmanned systems have several advantages, such as performing high-risk tasks (e.g., high radiation, contaminated areas or danger of explosion), more cost effective, and long-time survey and monitoring capability. Depending on the degree of human intervention on the robot’s decision (robot’s autonomy), they can be fully teleoperated (remote operated vehicle), semi-autonomous (aka supervisory control), or autonomous. Search and rescue robots, the most similar domain applicable to the radiation measurement scenarios, are normally either teleoperated or semi-autonomous [18].

### 3.1. Ground-Based Platforms

Ground-based surveys can be performed by either humans (foot-based) or vehicles [43]. Despite the fact that foot-based surveys (e.g., using handheld or backpack equipment) provide excellent spatial resolution, they require greater data collection time for large areas, which may be impracticable (e.g., due to radiation safe exposure limits) [17].

#### 3.1.1. Manned Ground Vehicles

Compared to foot-based surveys, vehicles, such as cars, trucks, or vans, can cover larger distances in less time (hundreds of km in a day) and are able to carry heavier payloads (large detection systems). However, their circulation is restricted to the existing road network and may be limited by the terrain typology (e.g., inaccessible places, like forests, cultivation fields), or other obstacles [17,43]. Moreover, to keep good spatial resolution, during the survey, the vehicle speed should not exceed 13 m/s [17].

Since the vehicle occupants may also be exposed to radiation risk, it is not the desirable method to be deployed in response to a scenario where the radiation intensity is high or is unknown (e.g., NPP accident or RN threat) [12,17].

#### 3.1.2. Unmanned Ground Vehicles

The use of UGVs in the dismantling of nuclear reactors, and in the aftermath of the accidents that occurred in the nuclear reactors of Chernobyl (1986) and Fukushima (2011), allowed them to operate in places with high levels of radiation, transport various sensors, and to perform measurements in real time [10]. However, limitations to the use of these vehicles, include inter alia, problems in terms of the sensitivity and performance of electronics in high radiation fields, difficulty in establishing communications, and reduced mobility of these vehicles/robots (e.g., descending and climbing stairs and overcoming some obstacles). Considering the tasks required to be performed in extreme scenarios, the use of UGVs will be limited, with human intervention (first responders and radiation task forces) being ultimately required.

#### 3.1.3. Wheeled Robots

Iqbal et al. [44] delivered a brief review of robotics in NPP, referring the use of robots with wheels (Figure 1a) or tracks or a combination of both (Figure 1b), with additional features, like stairs climbing ability and the integration of manipulators for inspection and maintenance. They concluded that the development of specific robots for reactor operations was not pre-planned; instead, it was a response to some need, thus reducing their efficiency for these tasks. The authors of Reference [18] also referred that most professional UGVs (no legged systems) available commercially are related to bomb disposal field and are normally propelled by tracks, run on batteries, and have manipulators for explosive device handling. A more extensive review of terrestrial robots for nuclear environments can be found in Reference [45]. A more recent example of the design and manufacture of a specific tracked robot for a nuclear accident scenario was described by Ma et al. [46] (Figure 1c).

Nagatani et al. [48] reported some tests of mobile robots inside a reactor building to assess the structural damage and dose levels in an emergency scenario. For that, the Quince robot was used, already with some capabilities for disaster scenarios (e.g., traverse bumps and stairs), and then retrofitted in order to respond to some specific issues: hardware reliability, communication (hybrid solution—wireless and long cable), and radiation hardness of the electronic components. Other features introduced into the robot were: A radiation sensor, a 3D laser range scanner, and a simple 2 degree-of-freedom manipulator. The system was later used in six real missions at the damaged reactor building of FDNPP.

Delemerito et al. [49] made a review of the current rescue robots (2014–2018).

#### 3.1.4. Non-Wheeled/Bio-Inspired Robots

Despite the fact that wheeled robots are normally chosen for search and rescue tasks, it is expected that bio-inspired robots starts to be used soon [18].

Many bio-inspired robots have now been developed, and some of them are already commercially available:Quadruped robots—can carry significant payload and may cross terrain with loose gravel or grass, as well as climb/descend stairs. Some examples are the SPOT robot from Boston Dynamics (up to 14 kg payload) [50], the ANYmal from ANYbotics (Figure 2a) [51], and the models Laikago/Aliengo/A1 from Unitree [52].Multi-legged robots—compared to quadruped robots, they have enhanced stability to walk in difficult and rough terrain. Examples of these robots are the small DLR Crawler from the Institute of Robotics and Mechatronics (Figure 2b) [53]; the Lauron series from FZI Research Center for Information Technology [54], which has adaptable behavior-based control and can also use the two front legs for manipulation purposes; and PhantomX AX Metal Hexapod MK-III from Trossen Robotics (Figure 2c) [55].Snake-like robots—due to their long and flexible body, they can move through complex environments and enter very confined spaces (e.g., pipes) (Figure 2d) [18,56].

Humanoid robots are also included in this category; however, they are still under development. They have the advantage of being able to manipulate objects and tools like humans (e.g., important for search and rescue situations) [18].

### 3.2. Underwater Platforms

Unmanned underwater vehicles (UUVs) can be divided into Remotely Operated Vehicles (ROVs) and Autonomous Underwater Vehicles (AUVs).

The exploration of inactive mines (in Europe, it is estimated that there are 30,000 inactive mining stations) may constitute a severe risk to humans. These environments are characterized by a network of tunnels (possibly flooded), in some cases with an unknown topography, and by the presence of metallic and industrial materials (e.g., cobalt, gallium, indium and rare earth elements). Thus, UUVs, particularly AUVs, are of great help to explore these sites allowing to extract topographic, geological, and mineralogical information [57]. Afterwards, this information can be used to determine if a mine can be drained and re-opened minimizing costs or if represents any risk of collapsing [58].

In the framework of the UNEXMIN European project, an underwater robotic system was developed for the autonomous exploration and 3D mapping of flooded and deep mines [59].

ROVs are also used in the inspection of nuclear reactor pressure vessels and other water-filled infrastructures (Figure 3), such as piping systems [60,61,62]. For that, robots normally carry water-resistance and radiation tolerance cameras and non-destructive evaluation sensors [61]. These inspections normally occur when the reactor is shut down for refueling and is becoming increasingly important with the aging of the worldwide NPPs [60]. In order to access confined spaces with complex structures, the robots should be: compact, highly maneuverable, untethered, body shape completely smooth (with minimal appendages), and radiation tolerant [61,62].

Specific underwater platforms are commercially available for the visual inspection and non-destructive examination of an NPP [63]. Water-resistant and radiation tolerant cameras are also available in References [63,64].

### 3.3. Air-Based Platforms

Aerial-based mobile detection can be divided into Fixed Wing, rotatory wing (single rotor or multi-rotor) and hybrid vertical take-off and landing (VTOL) fixed-wing aircraft. All these aircraft belong to a group called heavier-than-air platforms. Another important group is the lighter-than-air platforms which includes balloons and blimps (airships). A hybrid airship capable of transporting people and heavy cargo which is expected to spend only one tenth the fuel spent by a helicopter is under development by Lockheed Martin [65].

#### 3.3.1. Manned Aircrafts

Manned aircrafts (helicopters and fixed-wing aircraft) are normally used when a large area survey and a fast deployment/survey is necessary (e.g., a large-scale release of radioactive material into the environment after a nuclear accident) [66,67]. They also present a greater payload capacity compared to their unmanned counterpart—allowing them to transport large-volume radiation detection systems [67]. However, manned aircrafts are limited to minimum safety altitudes normally 152 m above ground level (AGL) in non-congestioned areas [68]. Moreover, the aircraft’s corresponding ground speed is a limitation factor for ground contamination measurements given the low spatial resolutions at reach. Despite the possibility of using helicopters to achieve lower altitudes, they also pose the problem of radiation exposure limits for the crew in high doses environments.

#### 3.3.2. Unmanned Aerial Vehicle

UAVs major technological acceleration within the field has been evident since the incident at the FDNPP in March 2011 [17]. Despite the use of only two UAVs (against seven UGVs) during the emergency in FDNPP [11], this event marked the first use of a SUAS, the Honeywell T-Hawk [69,70], a ducted fan UAV weighing 8 kg used for radiological surveys, structural damage assessment, and to foresee debris removal. The T-Hawk had a camera, a forward-looking infrared and a gamma dosimeter with time stamps readings and global positioning system (GPS) information. Even though this SUAS had autonomous vehicle and navigation capabilities its operation relied on the human-robot teaming. Since radiation intensity is inversely proportional to the square of distance and it is attenuated by the medium (air), it is important to consider SUAS due to their capability to fly at low altitudes and in close proximity to structures [69].

Missions where unmanned fixed-wing are superior compared to other unmanned platforms include radioactive plume tracking, sampling of airborne radioactive material, fallout mapping of large areas, and searching of unshielded sources, e.g., material out of regulatory control, both stationary and moving, from large areas.

Despite the fact that rotatory wing has advantages, such as hovering and VTOL capacity, they have relatively shorter operating endurance/range and payload capabilities.

VTOL aircrafts have several advantages as they can hover and need less space to launch and recover (do not need runway). They may include [71] multicopters/multi-rotors, such as quadcopters, hexacopters, or octocopters (e.g., quadcopter from Microdrones [72]), aerial robots (e.g., Honeywell T-Hawk [70]), single-rotor helicopter (e.g., Helicopter from UAVOS [73]), and fixed-wing hybrid VTOL, such as PD-1 from UKR SPEC systems (Figure 4a) [74]).

Despite their great potential for radiation monitoring near surface, to the best of authors knowledge, no work has been done using hybrid VTOL fixed-wing (complex system) nor blimps or balloons platforms. Due to their hover capability and long endurance (none or low fuel consumption), blimps can be used for environmental monitoring and inspection applications. An example is the project autonomous unmanned remote monitoring robotic airship, which explored many aspects related to the dynamics, control, and guidance methods of an airship [75,76].

A recently developed VTOL aircraft (2012) was the “plummet-proof” plane-blimp hybrid drone, also known as PLIMP (Figure 4b). To lift off, it uses both helium and rotational wings and, in the sky, can hover or maneuver better than a conventional blimp since it poses electric propellers for thrust and flight surfaces, like small wings [77].

In Table 2, a summary of the advantages and limitations of each air-based platform is provided.

It is worth noting that, while manned vehicles require a human being, normally, be exposed to radiological risks (among others), the UGV/UUV/UAV can work in a fleet or a swarm of vehicles mitigating that risk for the operators.

### 3.4. Challenges and Research

The use of unmanned systems in extreme environment missions and scenarios (e.g., nuclear disasters) still presents some challenges that need to be overcome [80]. In the same way, urban operation poses similar challenges, in particular, by the use of micro air vehicles (MAV), also known as drones (with dimension less than 1 m and speed lower than 10 m/s) [15], and by the use of SUAS [69].

Some issues must be overcome, such as:Communications [15,80]—use of payloads, like cameras (electro optic multi or hyperspectral cameras), light detection and ranging (LiDAR), and micro radio detection and ranging (RADAR), transmits high data volume and difficulties may arise due to limited bandwidth and possible interference or failure, particularly in operations "beyond line of sight". The latter may require separate frequency band or the use of satellite communications, which have higher latency, operational costs, and reliability issues. The requirements of higher bandwidth and secure communications are of special concern for robots that are based on distributed systems.Autonomous vs semi-autonomous—currently, robots need a high degree of human supervision and control, particularly in urban areas [80]. Due to low altitude flights (0.3–40 m) and proximity to urban structures (1.5 m) new challenges arise in vehicle and navigational autonomy. In such environments, five autonomous navigational capabilities must be considered: scan, obstacle avoidance, contour following, environment-aware return to home, and return to highest reading. In addition, the vicinity of buildings and other structures decreases the GPS satellite coverage. Therefore, autonomous capabilities should have the goal to increase the human skills, not to replace them, highlighting the human-robot teaming [69]. For indoor environments, GPS signal is not available.Data-to-decision process—improvement is needed in autonomous data analysis (visual and radiation data) for prompt use by mission commanders [69].Environmental sensors [80]—fast, cheap, and reliable sensors and associated electronics for real-time response are needed.Energy storage and management [15,80]—the current endurance of a battery powered rotary wing can vary between 10–60 min (depending on the payload).Weather conditions—in most cases, the operation of unmanned systems, particularly air-based platforms, are limited by adverse weather conditions (e.g., precipitation, wind, fog, haze, and pollution). The data collected by the sensors, the communications, and navigation systems might also be affected [15].Regulatory restrictions—safety regulations and operational procedures are necessary to avoid collisions of drones with ground obstacles (people and structures) and other aircrafts [15].Radiation damage [44,48,81]—when exposed to high radiation fields, the platform’s operational life is limited. This is due to microscopic damage caused by the radiation interaction with the platform materials. Therefore, it is important to predict the radiation damage in the platform materials and sensors to accomplish the planned mission tasks. Three ways are available to reduce the effects of radiation in critical components: increase the distance to source, reduce time exposure, and/or using shielding materials.Noise—low altitude and various UAVs might cause a significant level of annoyance (e.g., propellers rotation of a multi-rotor or airframe vibrations). Research may fall on both drone design (reducing the noise source) and flight paths [15].

Some examples of research projects and future technological breakthrough advances in the field encompass:Hybrid vehicles—e.g., commercially available unmanned VTOL fixed-wing, which does not need a runway (can hover) and has a flight time in the order of hours; or the use of a small agile ground robot carried by a multi-rotor type aircraft (Figure 5a) [80].Ducted fan drones—the ducted fan can produce more thrust than a open propeller with less power. This system protects the propellers from obstacles keeping also safe the surrounding people. Due to its inherent stability their use is being considered in radiological inspections [82]. A commercially available system is the platform AVID EDF-8 [83], which has a compact size (soccer-ball) and can navigate both indoors and in outdoor environments, particularly into narrow spaces. With a maximum payload of 0.45 kg can have an endurance up to 30 min.Bio-inspired robots—possible use of humanoid robots in nuclear power plants [44], an example is the research platform Atlas from Boston Dynamics [50], use of snake-like robots (Figure 5b) [84] as a sensing device for the inspection of the piping system of a nuclear power facility (research is needed in modular systems), and the use of flapping wing micro air vehicles for surveillance (e.g., homeland security) and monitoring missions [85].Cooperation between unmanned vehicles—Liu et al. [86] proposed an UAV which carries small ground robots to be deployed (e.g., by using parachutes and separation device modules) in the disaster area to collect detailed information. This way the ground robots can overcome possible obstacles and use the UAV as a communication relay to the ground control station (GCS).Swarm robotics—cooperation between multiple robots [44]. For example, a swarm quad-rotor robots for telecommunication network coverage area expansion in a disaster area [87].Cooperative navigation—for example, the use of a UGV to help improve positioning of a UAV in a GNSS-challenged environment [88].Computer vision [44]—improvements will help navigation and search algorithms to be more efficient.Robot learning and on board computing [44]—by using artificial intelligence and data fusion, robots might need minimum training to perform multiple tasks (e.g., deal with unwanted situations).Radiation damage—search for robot new constituent materials in order to protect the electronic devices [44]. Kazemeini et al. [81] studied the radiation damage of gamma-rays and neutron particles in electronic parts of a hexapod robotic platforms. A Monte Carlo transport code FLUKA was used to calculate the displacements per atom (DPA). Neutrons caused greater damage than photons and higher values of DPA/particle were obtained for silicon and copper parts of the actuators. To increase the operational life of the platform, different combinations of shielding (low and high atomic number materials) around the actuators were analyzed in order to have a trade-off between the applied shielding (payload) and the operational capabilities of the platform to accomplish the mission in the required time.

Other research fields referred in literature are: haptics, full autonomous operation, sensor technology, and powerful batteries [44].

## 4. Mobile Radiation Detection Systems

Since gamma rays can travel long distances in air (ranging from 65 m for 186 keV in ^235^U to 110 m for 662 keV in ^137^Cs), they are used for RN material detection and fingerprint. Neutrons also travel long distances in air, and normally result from spontaneous fission of heavy nucleus or generated in other nuclear reactions, like the absorption of alpha particles by certain nucleus. Beta and alpha particles can also be measured with mobile radiation detection systems; however, their short range in air (a few meters for beta and a few centimeters for alpha particles) makes detection difficult [89]. Therefore, in this article, it will be emphasized the detection of gamma and neutron radiation.

In the release and/or contamination by radioactive material, it is important to determine the radioisotopes present and the intensity of radiation in a given area. The field of radiation monitoring can fall into two distinct categories: (i) the location, identification, and quantification of a radioactive source; and (ii) the mapping of a contaminated area—aims at mapping the distribution of the radiation field over a pre-defined area [17]. This information allows quantification and scanning along space (geographical area in study) and evaluating, along time (study possible variations along the time), areas of risk, becoming a tool to support the decisions of governments and authorities. For example, the fast mapping of a contaminated area (e.g., post-disaster scenario) can provide valuable information for the safety of a task force and for the population evacuation plan.

This section is divided into five subsections: (i) brief review of the recent advancements in algorithms for the detection and search of radioactive sources, (ii) radiation detection and gamma spectrometry for ground and air surveys, (iii) gamma imaging, (iv) combination of neutron and gamma detection systems, and (v) dual particle (neutron and gamma) imaging systems.

### 4.1. Recent Advancements in Radiation Detection and Source Search Algorithms

Kumar et al. [90] reviewed the detection algorithms for radiation monitoring. According to the methodology, algorithms can be divided into: (i) true counting processing, (ii) spectroscopy processing for ground contamination monitoring, and (iii) plume tracking algorithms. Algorithms for true counting processing must consider corrections as height (e.g., altitude for airborne surveys), solid-angle, background, and detector efficiency (e.g., variation with energy), while, for spectroscopy processing (aids in source localization), one must consider Compton continuum elimination, de-noising, and stripping ratio (gamma spectra). For plume tracking, it is important to record the plume passage events.

The search of radioactive sources using mobile sensor networks was proposed in References [91,92], using Poisson Krigin techniques for the spatial distribution of radiation levels and source location, and in Reference [93], adopting a Bayesian framework and a sequential Monte Carlo for parameters estimation.

The authors of References [94,95] suggested the use of a set of measurements made by a mobile robot for the autonomous search of hotspots. While Huo et al. [94] used a search strategy based on a partially observable Markov decision process and a Bayesian framework for the source parameter estimation, Anderson et al. [95] used a recursive Bayesian estimation to predict the location and source intensity after each measurement. Future work is needed to consider objects attenuation, autonomous search in obstacle environments with multiple sources, and a cooperative search using land robots or a combination of land and air robots.

A method to integrate the position of a source into a grid map of the environment was developed, using a mobile robot capable of autonomous positioning [96]. The robot is composed by a radiation dosimeter, a LiDAR, and a mobile base with a odometer. Simultaneous localization and mapping (SLAM) technique is used for the grid map construction, and the estimation of the source parameters (position and intensity) is calculated with a Markov chain Monte Carlo algorithm (based on the radiation measurements). Despite the efficiency demonstrated by this method, some other effects must be considered for improved estimation of the source parameters, such as the wall’s contribution for radiation scattering, obstacles shielding effects, and background fluctuations. Future research is still needed in obtaining precise maps and in selecting optimal detection path. Since this method is not efficient for distributed sources localization (multiple sources), a way to improve it is by using a gamma camera.

The learning-based methods have been increasingly used in recent years. The authors of Reference [97] reviewed the machine learning algorithms in nuclear science and engineering highlighting the risks and opportunities of their application. Medhat et al. [98] proposed the use of an artificial neural network to identify radioisotopes in natural gamma sources and determine the uncertainty of the corresponding activity. Another promising use of artificial neural network was proposed by Jeon et al. [99], to reconstruct the Compton edges of plastic scintillator’s spectra for pseudo gamma spectroscopy, even with poor counting statistics.

### 4.2. Radiation Detection and Gamma Spectrometry

Radiation detection systems can be divided into three classes: gas-filled detectors, scintillation detectors, and solid-state detectors [100].

Due to their good sensitivity and energy resolution, scintillation detectors and solid-state detectors are used for radiation dose rate measurements and for gamma spectrometry. Gas-filled detectors, like the Geiger–Muller (GM) detector, are used to obtain the radiation dose rate. Some critical characteristics that measure the effectiveness of a radiation detector are: the energy resolution, correlated with the intensity of the light yield, counting efficiency which is related to the crystal ability to attenuate photons which depends on the incident gamma energy and is proportional to both the density and the atomic number of the material, and inherent dead time [16,43].

Scintillation detectors may be gaseous, liquid, or solid, organic (plastics, liquids) or inorganic. Their operating principle is based on the interaction of the incident radiation with the scintillator material and the conversion of the resulting light into electrical signals by a photodetector. Since inorganic scintillators exist in the form of a high Z crystal and a high light output, they are normally used for spectroscopy purposes (even for low energy radiation). In the same way, their high density offers good detection efficiency. Examples of inorganic scintillators are the alkali halides (e.g., NaI, CsI), oxides, such as the Bismuth Germanate (BGO), or lanthium halides (e.g., LaB, LaC).

A summary of some typical characteristics of common inorganic scintillators is given in Table 3. LaBr_3_[Ce+Sr] is a recent scintillating crystal featuring enhanced energy resolution (light yield) and photoelectron yield in comparison to the standard LaBr_3_ [101].

Since organic scintillators have low Z, Compton scattering is the main reaction, and photoelectric effect becomes dominant for low energies, typically below 20 keV. Their low density and low light output (e.g., SGC BC-400 plastic scintillator has a factor of four less light output than the inorganic scintillator NaI) makes them less efficient; however, this can be offset if large volumes of plastic scintillator are used (e.g., in RPM or waste monitors) due to their low price. Therefore, plastic detectors are normally used for gross counting gamma-rays above 100 keV. Other relevant characteristics are the ruggedness, very short decay time (a few ns), and the possible use for charged particles and neutrons detection. Despite the fact that both gas flow proportional counters and plastic scintillators have good efficiency for beta particles, are available in large volumes, are lightweight, and are available at lower price, the latter shows better efficiency (∼500 times than gas) for gamma-rays and a gain 10^3^ superior, i.e., a higher signal-to-noise (STN) ratio than proportional counters [108].

Phoswich detectors consist of a combination of scintillators with different pulse shape characteristics and are optically coupled to each other and to a single PMT (or PMTs), and they can also be used to: simultaneously measure multiple radiation types (alpha, beta, gamma, and/or neutron) and to measure low intensity of low energy photons in the presence of a high energy background, as is the case of uranium samples [109,110].

Another important characteristic of a scintillator detector system is the optical coupling of the output of the scintillating crystal to the associated light sensor. Table 4 displays some characteristics of light sensors that might be coupled to scintillating crystals, varying from the traditional PMT to the more modern SiPM.

When high energy resolution measurements are needed, semiconductor detectors should be considered. Their working principle is based on the creation of electron-hole pairs by the primary radiation or secondary particles. Applying an electric field, the collected charges form the output electrical signal. Silicon and germanium are widely used as semiconductor detector materials. The lower intrinsic detection efficiency of Si detectors (due to low atomic number and density) leads to their use in the detection of low-energy (soft) X-rays. For very high-resolution gamma spectroscopy, a high-purity Ge (HPGe) detector is normally chosen, which is considered the gold standard; however, they require cryogenic cooling to operate (heavier systems) [100]. High Z compound semiconductors as CdTe or Cadmium Zinc Telluride (CZT) ensures strong stopping power and high energy resolution gamma spectroscopy at room temperature. The widely used CZT can achieve an energy resolution better than 2.0% at 662 keV in routinely produced crystals (commercially off-the-shelf (COTS) detectors); however, they are expensive and their volumes are limited to a few cm^3^ [112,113]. The Medpix collaboration (CERN) developed highly pixelated fast read-out chips based on complementary metal-oxide semiconductor (CMOS) technology, namely the Medipix and the Timepix families. The possibility to combine these readout chips with various sensor materials (e.g., semiconductor sensors, gas-filled detectors, and microchannel plates) will continue to lead to the emergence of new hybrid “pixel detectors” for application in areas, such as: spectroscopic X-ray, gamma-ray, and particle imaging (including neutron imaging), space dosimetry, material analysis, and high-energy physics (e.g., particle track reconstruction) [114,115,116]. Recent developments in inorganic halide semiconductors, such as Thallium Bromide (TlBr) and perovskite halide crystals, have demonstrated high-efficiency and low-cost candidates for spectroscopic radiation detection at room temperature [117].

#### 4.2.1. Ground Survey

Following the FDNPP accident, it was necessary to measure the activity concentration of gamma-emitting radionuclides in environmental samples, like dust, soil, and pond areas. To collect dust sampling in the areas 20 km away from FDNPP, portable equipment mounted in cars were used. The samples were analyzed using semiconductor Ge detectors. For the ambient dose rate measurements, a GM counter, ionization chamber, and NaI(Tl) scintillation detector were used [118].

Since then, other car-borne surveys were undertaken to obtain air dose rate mapping of contaminated areas around Fukushima [119,120,121,122] or to determine the radiation background baseline at sites where a nuclear facility will be constructed [123]. For the data georeferentiation, a GPS was used, and, in some cases, also included were a PC and a router to send the data to a server via internet connection (for data backup or post-processing). The detection systems used vary from scintillation detectors NaI(Tl) and CsI(Tl), HPGe and Silicon Semiconductor Detectors. Recently, Prieto et al. [124] used two LaBr_3_ mounted in the car roof pointing out to each side (Figure 6a) for routine and emergency monitoring of large areas (mapping and plume tracking). To reduce the gamma-rays attenuation by the car structure, the detectors can be positioned outside the width of the car. The roof-mounted detector configuration is specially designed for radioactive plume tracking [122].

In order to be able to operate higher volume and heavy detection systems, Baeza et al. [125] proposed a van to obtain a fast response to an uncontrolled release of radionuclides (Figure 6b). The detection system consists of a 54.2 cm^3^ pressurized proportional counter (dose rate), a 5.08 × 5.08 cm NaI(Tl) scintillator and an HPGe semiconductor housed inside an iron shield for low background measurements of the aerosol samples collected. This mobile detection system also featured a meteorological station, a GPS (for position, altitude, and speed information), a near real-time data transmission by frames. The acquisition time varied between 30 s and 180 s, depending on whether it was an emergency simulated exercises or non-emergency events, respectively (trade-off between measuring error and spatial resolution). During the survey, a wide variability of dose rate values were reported, stemming mainly from difficulties in maintaining constant speed and the variability of soil composition. Temporary GPS failures were also reported, which lead to several data frames being not correctly georeferenced.

Walk surveys using backpack-mounted gamma-ray detection systems can be used to provide high-spatial resolution mapping of the distribution of contaminants in areas not accessible by vehicles (e.g., confined areas in urban environments) [126]. Some backpack systems described in the literature are CZT (260g) [127], CsI(Tl) (Figure 6c) [128], and a multipurpose detection system composed by two LaBr_3_(Ce) detectors (<6 kg) [129]. The latter also highlighted the advantage of integrating the detection system in other platforms, like in a fixed tripod and in vehicles, to reduce the uncertainty induced by each platform type. Nilsson et al. [130] investigated the performance of three different backpack systems, based on a LaBr_3_:Ce (9 kg), a NaI(Tl) (8 kg) and an HPGe (25 kg) at the site of a radioactive waste repository. Since the natural background at the site was low comparative to the self-activity of LaBr_3_:Ce, there was no significant improvement in using it instead of the NaI(Tl). The HPGe detection system showed the best performance (lower background, higher energy resolution, and intrinsic efficiency); however, its weight represents a limiting factor.

To avoid the radiation attenuation (up to 35%) induced by the operator body, a novel backpack equipment with two SIGMA-50 detectors placed on opposite sides and 1 m away from the operator was developed. The use of two detectors also allowed to reduce the survey time (increasing the detection efficiency) and the dose exposure experienced by the operator [66].

Relative to a handheld radiation detection system, Shokhirev et al. [131] used a 5.08 × 5.08 cm NaI(Tl) detection system (Figure 7a) with a battery for 7 h autonomy, connected via USB to an advanced processor for scintillators (APS), which in turn was connected via wireless to a smartphone (control and display interface). This detection system allowed real-time gamma-rays detection, isotope identification, and source localization by using the spin-to-locate (STL) procedure. The STL uses the radiation shielding of the operator’s body to estimate the source azimuth. Since this method depends on the attenuation of the gamma-ray flux by the body, its efficiency will depend on the energy of the emitting source.

In order to study and ascertain the site for the eventual construction of a nuclear reactor, Garba et al. [38] proposed a set of dose measurements at Kelantan State (Malasya). To accomplish this, a handheld detector (Model 19 microR survey meter, Ludlum) was used composed by a 2.54 cm (diameter) × 2.54 cm (length) NaI(Tl) crystal, to create a radiation map at a height of 1 m AGL. The higher activity areas were related to the geological soil composition (e.g., granitic origin).

Park et al. [132] developed a personal gamma spectrometer for homeland security and environmental radiation monitoring. The detection system consisted in a 3×3×20 cm^3^ Ce-doped Gd-Al-Ga-garnet (Ce:GAGG) crystal coupled to a SiPM (3 × 3 mm^2^) charged by a LiPo battery (3 h autonomy). Since the crystal is coupled to a SiPM, its total weight (battery included) is only 340 g, and it has a low power consumption (2.7 W). This scintillation material featured good technical characteristics: An energy resolution of 5.8% at 662 keV at room temperature, good sensitivity due to its high stopping power (material density of 6.63 g/cm^3^), and a good time response with a decay time of 90 ns shorter than CsI(Tl) and NaI(Tl).

Ozovizky et al. [111] developed an alarming PRD to be carried by a FLO. The detection system consisted of a cylindrical 14 mm (diameter) × 20 mm (length) CsI(Tl) scintillation sensor optically coupled to a SiPM. This sensor volume was calculated in order to meet ANSI 42.32 sensitivity requirements. In order to obtain the source direction, an array of 2 × 2 CsI(Tl) sensors (each sensor with 8 × 8 × 30 mm^3^ crystal coupled to a SiPM) was also considered. The CsI(Tl) crystal was chosen because of its mechanical properties, high density, good light yield and decay time. Its maximum emission wavelength (550 nm) also matches the SiPM quantum efficiency. The detector main features encompass low power consumption, good sensitivity (efficient light collection), low noise, pocket sized, lighter, automatic, robust, clear, and simple indication of radiation field presence and intensity. The use of SiPM allowed achievement of lighter detectors; however, it presented some drawbacks, like the poor (compared to similar detectors coupled to PMT) energy resolution of 15% at 662 keV and the temperature dependence.

Park and Joo [133] developed compact SiPM-based LYSO, BGO, and CsI(Tl) scintillators for homeland security applications. The energy resolutions obtained were 11.9% for LYSO, 15.5% for BGO, and 13.5% for CsI(Tl), using a SiPM array. Since these values are worse than using PMT-based scintillators, improvements are necessary in energy resolution and spectrum stabilization (temperature dependence). The reported advantages were related to their small size, low cost, high sensitivity, low voltage supply, and negligible magnetic influence in count rate.

Miller et al. [134] proposed a small, inexpensive and semi-autonomous mobile robot with a LaBr_3_ scintillator surrounded by a lead collimator that blocks gamma rays except those along the axial direction (Figure 7b). This allowed to perform a 180° horizontal scan rotating the detector (using a servo motor) and a directional profile of gamma radiation count rates is superimposed of the visual panorama. Zakaria et al. [135] also proposed a small robot with a GM detector to obtain the radiation map of an indoor environment using a predefined path. Despite the advantage of avoiding human exposure to radiation, mobile robots still need improvements, namely: Autonomous localization, navigation, mapping, exposure minimization, robot design (e.g., traverse irregular terrains), and communications.

A cooperative approach between a UGV and UAV was also proposed by Lazna et al. in order to combine the advantages of both platforms [136]. In this case, the UAV uses (e.g., multi-rotor) photogrammetric techniques to generate a 3D map of the region of interest (ROI) (terrain reconstruction), to assist the UGV (which carry a radiation detector) in the path planning to find a hotspot. The UAV can also carry a radiation detector to obtain a big picture in terms of possible hotspots. Compared to the separate use of platforms approach, this method is faster, more precise, and more reliable in finding hotspots in an unknown environment (with no a priori map of the terrain).

The concept of radiation detector connected to a smartphone to measure environmental radiation levels in real time was developed and implemented in the framework of the following projects: (i) mobile application for radiation intensity assessment (MARIA), using a GM counter [137], (ii) pocket Geiger (POKEGA), using a PIN photodiode [138], and (iii) mobile cloud system for rad monitoring (MCSR) consisting of a 36 cm^3^ CsI scintillation detector [139]. The fact that these detection systems are normally based on consumer-generated sensing (can be operated by citizens), sensor networking (by integrating collaborative missions to create a big picture), are lightweight and low cost equipment, makes them important tools to be used, for example, in the aftermath of a nuclear accident or to assess the effectiveness of decontamination efforts. MARIA project reported some constraints related to the survey speed and the distance detector-source is not constant and changes with user, and the difficulty in estimating only with the mobile phone and GPS) [140,141]. POKEGA project referred some limitations: the smartphone has an extremely low input gain and slow sampling rate (output signal from PIN photodiode is low and narrow), noise vibration susceptibility (incorrect readings), the energy consumption (smartphone battery insufficient), and the limited measuring range (sampling limitations) [138]. Considering the MCSR project, because the CsI detector needed a greater power supply, an external battery was used which led to a heavier system (3 kg). This scintillator had also an energy resolution of 13% at 662 keV. Since the efficiency of a real-time radiation monitoring by mobile detectors depends on the route planning, it was also analyzed this issue by considering two main factors: coverage and cost. As future work it is expected to optimize the surveys by solving the problem of route planning [139]. If low cost and lightweight detection systems can be disseminated among the population, this can provide valuable information to national authorities and response teams for fast decisions, not only in the vicinity of nuclear accidents but also at farther distances to consider the effects of the weather conditions (e.g., wind, rain or sea currents) or other non-intentional contamination spread out (e.g., transport by vehicles) [139].

The search for radiation sources in the framework of terrorism and malevolent acts or from radiological or nuclear accidents normally needs mobile robots to navigate without an environmental map. Lin et al. [142,143] proposed the use of an artificial potential field to navigate the robot through an unknown environment and a particle filter to estimate the gamma source position based on the radiation intensity measured by a radiation sensor (e.g., RedEye G).

Khan et al. [144] developed a detection system based on the dose rates of three GM detectors for robot coupling to be applied in radiological or nuclear emergency scenarios (even in high radiation fields), such as the search of a lost gamma source, radioactive contamination, and leakages.

Over time, nuclear reactors needed inspections, maintenance and repair to assure safe operation. Some reactor areas are too dangerous to be monitored by humans due to intense and high dose radiation fields; therefore, it is necessary to develop and implement remote monitoring solutions to perform these tasks at the minimum cost. Some detectors used in the inspection and maintenance of nuclear energy facilities and particle accelerators will be described below.

For the continuous operation of a pressurized heavy water reactor, it is necessary to refuel it in a day base by using a fuel exchange machine. Since this machine can get stuck to the pressure tube at a height of 9 m, it was necessary to develop a mobile robot with a telescopic mast to visually inspect it. Shin et al. [145] described the design criteria of the developed remotely operated robot. To avoid radiation failure of the system, the following solutions were adopted: (i) development of a radiation hardened camera able to stand radiation doses up to 1 kGy, (ii) development of a radiation dosimeter composed by a P-diode (dose-rate) and a pMOSFET (dose meter), both capable of measuring doses up to 1 kGy. The information about the total accumulated dose allowed estimation of the life expectancy of the robot, particularly the semiconductors that are part of the controllers, and (iii) use of a redundant emergency controller made by mechanical relays which are not affected by the radiation field.

For routine inspection and maintenance of an experimental fusion reactor, a remote handling system is used. The execution of these tasks is very time-consuming and expensive. Vale et al. [146] proposed the use of a multi-rotor UAV to accomplish the basic inspection of a nuclear reactor, in order to setup future planned inspections and maintenance. Some of the criteria proposed for the UAV were: modular system (more flexibility), flight autonomy, maximum payload, and cost. Regarding the sensors, it should include: An RGB camera with a wide angle of view, laser range finder (e.g., LiDAR) or a depth camera (e.g., Kinect), a thermal camera, and a GM. Promising results were obtained in an indoor scenario using weak sources. However, some limitations are expected when using UAV for the inspection of fusion reactors (indoor), namely the high temperature and the high dose rates [147]. This concept is also valid for other scenarios, like the inspection of contaminated areas in fission reactors (e.g., leakage detection), storage areas (nuclear sources), or reactor accidents.

In order to choose a scintillator for use in large fusion facilities, Sibczynski et al. [103] compared the characteristics of several detectors (LaBr_3_:Ce, CeBr3 and GAGG:Ce, CsI:Tl, and NaI:Tl). They concluded that the best candidates for the gamma-ray spectrometry and X-ray emission measurements in high-temperature plasma experiments are the LaBr_3_:Ce and CeBr_3_ scintillators because of their high count rate capability. Since the detectors will be used in strong magnetic fields, coupling these detectors to semiconductor photodiodes, such as SiPMs, multipixel photon counters (MPPCs), and avalanche photodiodes (APDs), instead of the PMTs (susceptible to magnetic fields), is also considered.

Celeste et al. [148] described the development of a handheld radiation survey meter (B-RAD) to be operated inside a strong magnetic field, such as in the experimental areas of the CERN Large Hadron Collider and its ATLAS detector. In order to operate in such scenarios, some minimum requirements were established, like: insensitivity to magnetic field (up to 1 T), dose rate between 0.1 μSv/h (ambient background) and a few mSv/h, 60 keV threshold, and an energy range up to 1.4 MeV (the average energy of gamma-rays from residual radioactivity in particle accelerators is 800 keV). The U.S. Department of Homeland Security published, in 2017 [149], a list of PRDs and spectroscopic personal radiation detectors, along with their specifications; however, only four out of 23 complied with the requirements previously mentioned. The B-RAD consists of a LaBr_3_ scintillation crystal coupled to an array of SiPM. The selection of this crystal material was related to: its fast decay time in order to account for high dose rates (on the order of several hundred thousand cps), high density, good temperature stability, a high light yield, and a good energy resolution (∼3% at 662 keV). Five crystal geometries were tested in order to select the one that best fits the SiPM (4 × 4 mm^2^) and to obtain the trade-off between sensitivity (crystal size) and the capability to account for dose rates greater than 1 mSv/h. An important finding was related to the increase in temperature which lead to a significant reduction (depending on the crystal) on the count rate down to about 50–70% of the initial value (maximum value). This temperature dependence of the detector response can be explained by the variation of the crystal light yield and by the drift of the breakdown voltage of the SiPM with temperature (estimated in 38 mV/°C). To compensate this, a circuit (bias voltage adjustment) to stabilize the SiPM gain was implemented. Finally, B-RAD can be also used for industry, particularly scenarios which require dose rate measurements in the presence of high intensity magnetic fields, as is the case of scrap metal handling or containers handling using lifting magnets, and in medicine applications as surveys around positron emission tomography/magnetic resonance imaging (MRI) scanners or in the recent technology of medical electron linear accelerators coupled to MRI equipment for image-guided radiation therapy [148].

#### 4.2.2. Airborne Survey

After the Fukushima accident, there was a serious lack of information, not only regarding the identification of the radioactive materials released but also their distribution as a consequence of the plume’s fall-out in neighboring regions [17]. Traditional airborne (manned aircrafts) measurements of radiation intensity, dose, and corresponding spatial distribution were performed at altitudes of 150–300 m. First, measurements were done by using NaI scintillation detectors to obtain an estimate of the ambient dose rate at 1 m height within an area of 80 km from the FDNPP [118].

Sanada et al. [150] described the use of manned helicopters to measure the ambient dose-rate and the deposition of radioactive cesium in the vicinity of FDNPP using large NaI detectors mounted either inside or outside the platform. The helicopters have the advantage of reaching difficult areas for humans, like forests and paddy fields [151]. Due to regulatory restrictions (manned aircrafts cannot fly below 150 m) and to avoid contamination and exposure risks to the crews, the flying altitudes were above 150 m, which resulted in poorer spatial resolution. Therefore, to obtain the necessary detection sensitivity, heavy, high volume, and expensive radiation detectors were necessary [152].

A more recent work reported on the use of several NaI detectors and a LaBr_3_ installed in a manned helicopter to determine the influence of natural radionuclides (radon progenies, ^214^Pb and ^214^Bi) in aerial radiation monitoring for a better estimation of the deposition of artificial radionuclides in the vicinity of Fukushima [153].

Castelluccio et al. [154] described a detection system housed in a manned fixed-wing aircraft for air-sampling and ground contamination measurements in large areas, e.g., the aftermath of a nuclear or radiological accident. To accomplish that, a Sky Arrow 650 aircraft was modified (the aircraft nose), in order to accommodate the air sampling unit. To analyze the aerosol samples (collected in the filter), a GM and a BGO scintillator (1 cm^3^) next to the sampling line is used. However, since the BGO scintillator has poor energy resolution, an HPGe semiconductor (nitrogen cooled for 4 h) with a 60 mm (diameter) × 35 mm (length) crystal size to radionuclides identification is also used. To measure the ground contamination (dose rate estimate), a large volume NaI(Tl) scintillation detector (400×100×100 mm^3^) weighing 17.5 kg is used. One important condition for correct air sampling measurements is to guarantee the isokinetic condition, ie, the air streamlines and the airspeed must not be perturbed by any part of the airplane.

Due to possible dose risks to the crew, in an emergency situation, the manned airplane can only measure environmental contamination in a far field situation, while near field situations can only be handled using an UAV or a network of UAV.

The FDNPP accident marked the first use of UAVs to obtain the distribution of air dose rate and radioactive cesium in the ground (^134^Cs and ^137^Cs) nearby the nuclear power plant. The use of unmanned helicopters allowed to carry a more reasonable payload weight and also perform measurements at lower altitudes (50–150 m) and speeds (8 m/s) than manned aircrafts, which allowed improved spatial resolution [78,151,155,156]. While Towler et al. [155] used an NaI(Tl) scintillating detector for the mapping of the radiation distribution and localization of hotspots, Sanada et al. [78,151] and Nishizawa et al. [156] used three LaBr_3_:Ce (6.5 kg) to measure the distribution and variation along the time of the air dose rate and a single LaBr_3_:Ce to determine the ratio ^134^Cs and ^137^Cs, respectively. The choice of LaBr_3_ was related to its higher efficiency and energy resolution compared to NaI crystals, allowing to distinguish the energy lines of the radioactive ^134^Cs (796 keV) and ^137^Cs (662 keV). Towler et al. also used other sensors as a stereovision system to generate terrain maps and a gimbal camera with laser rangefinder to geo-locate points of interest.

After the Fukushima accident, several lightweight (50–500g) spectrometers started to be available as commercially off-the-shelf (COTS), like CZT semiconductor, CsI, and CsI(Tl) scintillation detectors [18]. In Table 5, some characteristics of COTS detectors are presented.

Recently, Saint Gobain made available compact SiPM-based scintillators, such as: (i) NaI(Tl) with an energy resolution between 7.5–8.5% at 662 keV (depends on the crystal size), (ii) NaIL (NaI(Tl+Li) a dual mode detector for gamma radiation and thermal neutrons, and (iii) LaBr_3_:Ce with an energy resolution of 4% at 662 keV for a crystal with 3.81 cm (diameter) × 3.81 cm (length), and (iv) enhanced LaBr_3_:Ce with an energy resolution of 3.5% at 662 keV for a crystal with 3.81 cm (diameter) × 3.81 cm (length) [157]. They feature temperature gain compensation, low voltage operation (5 V, with a power consumption <150 mW), and are tested according to ANSI N42.34.

MacFarlane et al. (2014) [152] implemented a compact, lightweight, and small volume CZT detector (1 cm^3^ GR1, Kromek) coupled to a small UAV (multi-rotor) for monitoring, assessment, and mapping radiation anomalies. The payload consisted in a gimbaled system (to ensure sensors are perpendicular to the ground surface) with a CZT and a single-point laser rangefinder (AR2500, Acuity) to obtain high precision heights. A similar setup was used by Martin et al. [158,159,160] (Figure 8a) to obtain high-resolution radiation mapping of legacy uranium mines, verify the effectiveness of various remediation methods, and to investigate contaminant migration in FDNPP post-disaster scenario including 3D mapping (using a software to visualize 3D spatial data). Due to the flight characteristics of the UAV (heights between 1–15 m and speeds between 1–1.5 m/s), it was possible to monitory elevations of land and infrastructure, allowing to measure not only the radiation fields but also to identify the radionuclides present. In general, this detection system showed the following advantages: low operation and maintenance cost (compared to fixed-wing), fast deployment, and accomplish autonomous missions. However, some limitations were identified, like the platform low autonomy (30–35 min), the strong weather dependence, and the small volume size sensors (payload limitation), compared to high altitude fixed-wing platforms. As future work, the use of a 3D-scanning LiDAR system (32 lasers) is referred for monitoring and contamination mapping of structures, such as buildings and waste material storage.

Recently, Connor et al. [164] used a SIGMA-50 CsI(Tl) scintillation detector (with SiPM) coupled to a multi-rotor to assess the remediation efforts (contaminant distribution) in a Fukushima waste storage area (Figure 8b). Despite the better energy resolution of CZT (used in previous works [158,159,160]) compared to CsI(Tl) (2–2.5% vs <7% at 662 keV), the latter detector was chosen because of its larger volume (32.8 cm^3^) and correspondingly higher efficiency, which allowed obtaining the mapping of lower radiation intensity areas.

For high resolution mapping at NORM sites, particularly decommissioned uranium mines, a lightweight gamma spectrometer coupled to a multi-rotor was proposed [140,158]. While Martin et al. [158] proposed a CZT to scan the area at 5–15 m AGL and speed 1.5 m/s, Borbinha et al. [140] suggested lower altitudes and speeds (1–2 m height and 0.2–0.23 m/s) using a GM for a first monitoring (hotspot identification) and only after the use of a CZT for hotspot inspection (energy spectra obtained at 10–20 cm from the ground—platform landed). The latter is related to the FRIENDS project, which consists of using a combination of sensors coupled in a fleet of drones (autonomous and cooperative navigation) with the goal of monitoring and mapping in real time areas with high concentrations of NORMs. Following this project, Brouwer et al. [141] explored different approaches to estimate the number of sources, intensity, and their location. It was concluded that the maximum likelihood algorithm provides the best results, and the application of a collimator into the radiation sensor improves the source localization; however, it has a negative impact on the payload and STN ratio. Future research is required on path planning for survey time optimization.

To solve the lack of software and hardware specific for radioactive source detection, a CZT coupled to an UAV (multi-rotor) and a laser altimeter for precision height measurements was implemented, as well as a software to manage the communication and data was storage [165]. For example, depending on the signal latency, the RIMAspec software can commute between the available links (private radio networks, Wi-Fi, and 3G/4G) to ensure best data transmission to the ground station. Since the uncertainties arouse mainly from the distance to the source and the low statistics, in the future, the CZT will be replaced by a higher efficiency 5.08 × 5.08 (cm) NaI scintillator.

In order to increase the situation awareness of the operator, a visuo-haptic augmented reality interface was proposed for the teleoperation of a multi-rotor equipped with a CZT detector for the localization of nuclear sources in outdoor environments [167].

To characterize the soils contaminated by ^241^Am and ^152^Eu (anthropogenic causes), Falciglia et al. [168] proposed another compact and lightweight (180 g) gamma-ray spectrometer, a semiconductor CdTe detector with an active area of 25 mm^2^ and a thickness of 1 mm, with a energy resolution <1.5% full width at half-maximum (FWHM) at 122 keV, coupled to a multi-rotor. They studied the effects of the flight parameters, such as the height, inclination of the detection system relative to the soil, and the detection time in the detection efficiency and minimum detectable activity (MDA).

Cai et al. [169] proposed a small ducted fan UAV (AVID EDF-8) to carry a lightweight radiation sensor (Teviso RD3024—PIN Diode) for radiation detection (gamma and beta) and mapping in a nuclear emergency scenario. It was highlighted that, for critical time tasks, a path planning algorithm should be developed, which could change the search for minimum or maximum dose values (instead of an exhaustive search pattern), e.g., to find a safe path for rescues or to find hotspots.

Salek et al. [166] showed the possibility of using a multi-rotor to carry a relatively large volume detection system to environmental monitoring, particularly uranium anomalies near the village of Třebsko, Czech Republic. The detection system is composed by two 103 cm^3^ BGO gamma-ray spectrometers (Figure 8c), with a total weight of 4 kg (batteries included), and is characterized by an energy resolution of 13.6% at 662 keV. The combined detectors have a total sensitivity of 160% compared to the standard 350 cm^3^ NaI(Tl) crystal and each BGO has about 300 times more sensitivity than a 1 cm^3^ CZT detector. With a speed of 1 m/s and heights between 5–40 m, it was possible to collect the same number of counts per unit distance as a standard airborne survey (manned aircraft). However, limitations, like the flight autonomy of only 16 min (related to the payload weight) and speed, make the mini-airborne gamma-ray spectrometry impracticable to regional surveys. The authors also argue that, despite their good energy resolution, the Labr_3_(Ce) scintillation detector and the semiconductors CZT or CdTe described above are not appropriate to measure low radiation fields (comparable to background values), the first one because of the crystal intrinsic activity by ^138^La and the other two due to their small sensitivity (small volume sensors).

Lüley et al. [8] also proposed a detection system coupled to a multi-rotor for dose rate measurements, air sampling, and radiation mapping. One GM (0.5 kg) and an air sampler module for aerosol collection (<1.5 kg) were used. The devices main features are: Modularity (detector and air sampler in modules), customized control software, hibernate capability (system activates when necessary), easy to clean (if contamination occur), easy to integrate on different UAVs, possible parallel operation of UAVs to improve map efficiency, and online radiation map functionality (data available in real-time).

One way to overcome the flight time limitations (range) and the low speed of multi-rotors, as well as the lack of spatial resolution on the data collected by detection systems coupled to manned aircraft (e.g., flight altitude regulatory restrictions), is by the use of an unmanned fixed-wing as the transport platform.

Connor et al. [67] presented a fixed-wing UAV solution for mapping large contaminated areas, like nuclear power plant post-disasters (Figure 9a). The take-off weight is 8.5 kg (1 kg payload) and features an autonomy greater than 1 h. The payload consisted in two SIGMA-50 CsI(Tl) scintillation detectors. Due to the survey speed of 14–18 m/s and low altitude measurements (40–60 m), it was possible to achieve dose rate maps with a high spatial resolution (20 m pixel^−1^). A laser-range finder on board was used to measure the height above ground; however, the tall vegetation canopies prevented the laser beam to reach the ground (wrong height results). Therefore, a digital elevation model was used to directly correct all measurements to obtain dose-rate at 1 m AGL. Some technical features of the platform system are: the possibility to hand-launch and recover the platform by parachute, flight autonomy greater than 1 h that allows launching the platform from a safe place, accomplishing the mission, and returning back, and navigation autonomy (fly according to pre-planned waypoints). Some of the drawbacks reported were: the saturation of the detectors due to the high doses encountered nearby the “Red forest”, as well as the weather influence in UAV flights (rain and wind)—sometimes the local wind variation led to ground speeds of 25 m/s (tailwind), despite target speed selection of 14–18 m/s. From the survey experience, the time response of the detection system will be improved to accomplish higher gamma fluxes, by replacing one SIGMA-50 by one CZT detector (GR1, Kromek). The use of CeBr_3_ and LaBr_3_ scintillators is also anticipated in the near future—due to their excellent energy resolution and optical yields, even for small sensor volumes.

The development of radiation sensors in unmanned fixed-wings aircraft was also explored by Pollanen and Smolander [170]. Two UAV were equipped with radiation detectors:Mid-sized Ranger aircraft (Figure 9b)—with an autonomy of 5 h and flight speeds from 100–220 km/h, a maximum take-off weight (MTOW) of 270 kg, and a payload capacity of 40 kg. In this platform, a GM counter for external dose rate monitoring was installed, a 15.24 × 10.16 (cm) NaI(Tl) scintillation detector used for radioactive plume localization, and a 5 × 5 × 5 mm^3^ CZT detector housed inside the sampling unit (this detector also accounts for the possible saturation of the large NaI scintillator).Patria MASS mini-UAV (Figure 9c)—with an autonomy of 1 h at cruise speed of 60 km/h and an MTOW of 3 kg, it can transport payloads up to 0.5 kg. A cylindrical CsI detector with 38 mm (diameter) × 13 mm (length) crystal was used in the fuselage, which revealed a poor energy resolution (12% at 662 keV), and a radioactive particle air sampler was mounted above the aircraft.

UAVs can operate in both manual and automatic fly modes, and the measurement data can be sent in real time to the ground station.

Lowdon et al. [43] used the GEANT4 simulation tool to analyze the characteristics of CsI(Na), CsI(Tl), LaBr_3_, and cerium-doped lutetium-yttrium oxyorthosilicate LYSO(Ce) scintillation crystals for use in airborne environmental radiation monitoring, particularly UAVs. They concluded that LaBr_3_ is the best candidate due to its detection efficiency and energy resolution. However, due to the self-activity of this crystal and the corresponding increase in the STN ratio at low energies, an alternative, such as the CeBr_3_ crystal, must be analyzed in the future, as well as a better way to shield these crystals against atmospheric exposure (hygroscopic behavior).

The fast search for radioactive sources in security (e.g., lost or stolen source) and safety scenarios (e.g., nuclear disaster mitigation) can be achieved using UAV. Despite the fact taht a fixed-wing UAV can be used for fast surveys, these platforms normally require higher source activities (to be detected) and need a runway. On the other hand, a small and low-cost multi-rotor platform can be used for source localization and contour mapping in all-terrain and confined spaces (e.g., mountain or urban areas) and for lower sources activities [171]. Baca et al. [172] implemented a 14 × 14 mm^2^, 256 × 256 pixel matrix CMOS detector, known as Timepix, coupled to a multi-rotor. Despite the implementation of a single Timepix detector to source localization and mapping (e.g., in security and nuclear disaster mitigation), the development of a Compton camera made by two Timepix3 detectors coupled to MAV in order to localize weak gamma-ray sources is expected. The high mobility of an MAV allied to the lightweight Compton camera data will allow full autonomous localization of radioactive sources.

The use of a UAV for multiple source localization and contour mapping (e.g., for fast emergency response) was also proposed by Newaz et al. [173,174]. When considering multiple sources, each of them contributes in a cumulative manner for a given hotspot, which differs from the situation in which each source is considered individually. To agile the process, a region of interest (ROI) is first considered and then the source localization is based on two methods: Hough transform and the variational Bayesian.

A multiple formation of UAVs was proposed for cooperative source seeking and contour mapping. While Han et al. [175] proposed the use of low-cost UAV formation (e.g., fixed-wing aircrafts) in four different scenarios, which included the study of a decentralized formation strategy to increase its robustness in the case of a communication failure (with the GCS) and different types of formations (circular and square), Cook et al. [176] considered a circular formation of three multi-rotors to discover the radiation intensity gradient for low altitude and clustered environments (only feasible with this kind of platforms). The latter used a plug-and-play concept for the integration of the CZT detectors into the platforms. For the urgent detection of nuclear radiation, a multi-UAV system has several advantages compared to a single UAV system, mainly related to the use of low-cost, small, and lightweight platforms, which translates in more safety, flexibility, and endurance [177].

### 4.3. Gamma Imaging

Gamma cameras can be divided accordingly to three different principles: Pinhole, Coded-aperture, or Compton scattering (also referred to as Compton camera) [19,178].

Pinhole cameras (Figure 10a) are based on thick and heavy collimators (Pb or W) to prevent the entry of gamma-rays outside the detector field of view (FOV). In order to achieve good angular resolution, the pinhole geometric area must be as small as possible, therefore reducing the detection efficiency [22]. These collimators also make the detection system too heavy and difficult to handle [19].

Compton cameras (Figure 10b) use the kinematics of Compton scattering to estimate the source location (Compton cone), which can be derived from the angle made by the first gammas interactions in the scatter array (first layer of detectors) and in the absorber array of detectors [178]. Since this type of camera do not use any collimation or coded mask, it features a wide FOV [22].

Coded-aperture cameras (Figure 10c) work with a passive or active mask to modulate the incident gamma flux that reaches a position sensitive detector to generate gamma images [179].

According to Reference [19], gamma cameras are not suitable for the rapid detection of weak sources, being more suited to be used in non-dynamic scenarios with large amounts of radioactive materials.

Despite the fact that gamma cameras are a very powerful tool to identify hot spots or map regions with concentrations of radioactive materials, as in the aftermath of a nuclear accident, their images are devoid of source–camera distance information, as well as depth information of radionuclides embedded in materials (e.g., sand, concrete). Some methods may be used to determine these missing information; for example, Iwamoto et al. [180] proposed methods to obtain the depth information in materials analyzing the spectra of the radionuclides and checking the ratio of scattered to direct gamma-rays.

The use of gamma cameras for the autonomous mapping and hotspot localization is limited by their long acquisition time and poor angular resolution. In order to reduce these sensor limitations, Ardiny et al. [181] implemented some exploration algorithms based on two exploration approaches, the behavior-based and multi-criteria decision making (MCDM). For the analysis of the influence of FOV and angular resolution, parameters in the localization of a few sources (e.g., one or two), as well as the number of stops the gamma camera needed to create a radiation image, were considered. It was concluded that MCDM features the best results especially in complex environments. Moreover, a gamma camera with higher FOV and poor angular resolution (compared to a camera with poor FOV and good angular resolution) would be preferable to explore more areas over a given time. It was also referred the importance of multi-robot systems to improve hotspots localization due to their advantages, like information sharing (faster surveys), robustness, and fault tolerance.

#### 4.3.1. Compton Cameras

Sinclair et al. [182,183] developed a SiPM-based Compton telescope for safety and security (SCoTSS) to be used by nuclear emergency response and security teams. The SCoTSS was the first Compton gamma imager based on solid scintillators coupled to SiPM, performing as a directional survey spectrometer and a source localization equipment. It consists of pixelized layers of crystal scintillators made by two scatter layers of CsI(Tl) scintillating crystals coupled to SiPMs and a absorber layer of NaI(Tl) scintillating crystals coupled to PMTs. The small and lightweight SiPMs allow to reduce the scattering of gamma-rays in the dead material. Since this camera is modular, it can be used as a handheld or backpack instrument (single module with 4 × 4 array of crystals) or installed in a mobile platform (e.g., truck) using the 3 × 3 module configuration. This latter configuration can achieve a localization precision better than 2° for an integration time of 3 s using a 1 mCi Cs-137 source at a 10 m distance. This performance anticipates their use for imaging in motion. For field measurements, it will be necessary to account for the control gain variation of SiPM due to their temperature dependence.

A prototype Compton camera carried by an unmanned helicopter (Figure 11a) for localizing radionuclides was developed by Jiang et al. (2016) [178]. The gamma camera was composed by two arrays of Ce:Gd_3_(Al,Ga)_5_O_12_ (Ce:GAGG) crystals coupled to SiPMs (scatter array) and to APDs (absorber array). This work was followed by Shikaze et al. [184], who tested an improved version of the gamma camera nearby the FDNPP site. The choice of Ce:GAGG crystals was mainly due to their low cost (compared to NaI and LaBr_3_), sufficient energy resolution to discriminate the ^137^Cs and ^134^Cs peaks (between 600–700 keV), high light yield, and good Compton efficiency. Compton images were obtained while the helicopter was hovering, e.g., during 5 min at a height of 10 m was enough to obtain the gamma-ray image of a 400 m^2^ area with a dose rate between 5–12 μSv/h. The fusion technique of the 3D scene with the contamination distribution is amongst future improvements needed. The application of this system for the fast evaluation of decontamination efforts, hotspots search, and assessment of the distribution and mobility of ^137^Cs over large areas of difficult access in Fukushima (e.g., forests, riverbeds, residential areas) can be anticipated.

The first aerial demonstration of a Compton camera carried by a multi-rotor was achieved by Mochizuki et al. (2017) [185]. The compact Compton camera also made with Ce:GAGG scintillating crystals (1.9 kg) has the following advantages: high density, high light yield, non-deliquescence, and no self-activity. This imaging method allowed faster (factor of 10) surveys compared to foot-based, measurements in places of difficult access, and better spatial resolution compared to unmanned helicopters. The most important limitation is the short flight time (10–20 m). A significant impact of the altitude accuracy in the resultant images was also reported.

The use of multi-rotors UAV to carry lightweight Compton cameras was also developed by Sato et al. (Figure 11b) [186]. Unlike helicopters, these platforms are very useful in obtaining remote measurements of radioactive contamination in narrow areas, like the interior of the FDNPP buildings. The gamma imager consist of a two stages (scatterer and absorber) of Ce:GAGG scintillation crystals coupled with an MPPC (1.5 kg) mounted in a gimbal in order to maintain the FOV angle. The flight autonomy is only 16 min, and the time required for an image reconstruction is approximately 550 s (almost 10 min) hovering at 9 m height from the sources with tens of *μ*Sv/h. Since the contamination can be anywhere in a building, a 4π Compton camera is being developed. Sato et al. [187] also performed radiation imaging while the platform is moving by measuring the self-position and posture of the camera in each detection (using time stamps), which allowed extraction of the direction of the Compton cones. Replacing the gamma imager by a 3D-LiDAR, it was also possible to do a second autonomous flight to obtain the 3D topographical measurement and to create a 3D radiation distribution map.

A volumetric (3D) Compton imaging (VCI) system to be coupled on a mobile platform was also developed to improve source searching and mapping of unknown environments in nuclear security and safety scenarios [188,189]. This equipment is based on the data fusion of the Compton imager with a Kinect sensor (RGB images) for real-time tracking and 3D scene reconstruction (using SLAM techniques). While Barnowski et al. [188] used a VCI based on HPGe detectors, Haefner et al. [189] implemented a handheld high-efficiency multimode image, based on a two-plane active-mask configuration, which allowed the use of both coded-aperture and Compton imaging modes (total weight 3.6 kg). Haefner et al. also reported some important features, like: 3D scene data fusion to improve scene geometry (e.g., RGB data and depth information from Kinect) with gamma-ray image reconstruction from several sources, improvement of spatial resolution and mitigation of the 1/r^2^ intensity reduction by bringing the gamma imager closer to the objects, use of RGBD-SLAM algorithm (in which RGB stands for the visual data and D for depth obtained by the Kinect sensor) to simultaneous create 3D model of the environment and to track the detector position and orientation, and list-mode operation using maximum likelihood expectation maximization (ML-EM) method to gamma-ray image reconstruction. Three-dimensional scene data fusion is a valuable tool that can be used to localize sources hidden inside an object and can be useful for the detection and mapping of other types of particles (e.g., neutrons). A drawback comes from the fact that Kinect sensor only works in indoor scenarios since it depends on active IR light. The authors refer the possible use of visual cameras to create 3D models of either indoors or outdoors environments.

The use of a moving Compton Camera imaging robot was explored to obtain 3D reconstruction of radiation image for source recognition (e.g., leakage detection and disaster recovery) by Kim et al. [190] and Cong et al. [191]. The former used the SLAM method to estimate the robot trajectory (pose of the detector) and for the 3D reconstruction used the ML-EM based on all measured data and estimated poses. This method allows estimation of the source position even if the source is blocked by an obstacle (wall or door). The latter used a new method called cross-section outline at half maximum to assess the precision of the reconstructed image.

Sato et al. [192,193] also tested a compact Compton camera (scatterer and absorber made by Ce-doped GAGG) mounted on a crawler robot for radiation distribution measurement inside the FDNPP buildings to help decommission tasks. The combination of a gamma-ray imager and a robot is especially important in wide areas and high radiation fields (e.g., inside the FDNPP). A virtual reality was used to display data of the 3D data of the radiation distribution to workers to avoid excessive radiation exposure. The 3D structural model of the working environment can be obtained by photogrammetry software or by using 3D-LiDAR (can measure the distances to the objects), while the position and posture of the imaging system can be obtained by a SLAM function.

Kataoka et al. [22] described the development of a novel handheld Compton camera with only 1 kg and 10 cm^2^ size (Figure 12). The camera is composed by 10 mm thick plates of 50×50 mm^2^ Ce:GAGG as scatter and absorber, as well as features a wide field of vision (180°) and an improved sensitivity (approximately 1% for 662 keV gamma-rays). Using the 3D position-sensitive scintillators together with large-area monolithic MPPC arrays, it was possible to reconstruct a gamma image in 30 s (integration time) for a ^137^Cs source with approximately 6 μSv/h. The resultant angular resolution was 14° (FWHM).

The first commercial portable Si/CdTe Compton camera was manufactured by Takeda et al. [194] (ASTROCAM) and tested for hotspot detection and evaluation of decontamination effectiveness nearby Fukushima (in the 20 km zone). The camera weights 10 kg and the efficiency can be improved for a specific application by changing the number of detectors (scalability). In this camera, Si is used as the scatter, and CdTe is the absorber, wherein the former allows a smaller Doppler broadening effect, and the latter a good absorption, due to the high atomic numbers (48 and 52) and high density (5.8 g/cm^3^). The energy resolution is 2.2% at 662 keV, with an angular resolution of 5.4° (FWHM), and the exposure time used during the tests was 30 min. Particular attention must be paid when intense hotspot is near the camera because it can be difficult to detect the furthest of hotspots. To avoid this situation, the camera must be raised to some high point; doing this, it is possible to observe hotspots 20–30 m away from the camera.

Another portable semiconductor-based Compton camera was developed by Wahl et al. [195] for spectroscopy and imaging in nuclear power plants. The Polaris-H is a 4π Compton camera based on a 3D-position-sensitive pixelated CZT detector (energy resolution better than 1.1% at 662 keV) with a total mass of 4 kg and an autonomy of 5 h.

#### 4.3.2. Coded-Aperture Cameras

A 3D stand-off radiation detection system (SORDS-3D) was developed by Penny et al. (Figure 13a) [179]. Housed in a cargo trailer (total weight 1200 kg) it is composed by a central detector array (with 37-element) of 5 × 5 × 50 cm^3^ CsI(Na) scintillation detectors and a uniform redundant array coded-aperture masks on the left and right sides of the detection system to modulate incident radiation. At either end of the detectors, there is a PMT, and the light ratio between them is used to locate the interaction point of the incident gammas along the detector’s length (allows to determine source’s elevation). An energy resolution of 6.13% at 662 keV was reported. Therefore, the detection system can locate in 3D and identify compact radioactive sources with background rejection in real time. Side-facing video cameras are also used, as well as a GPS, environment controls, and a generator. At speeds up to 95 km/h and for 100 m stand-offs, detection and localization of sources (including source elevation) on the order of mCi was reported. Despite being a large-area detection system, its detection capability of compact sources at greater distances is normally limited by the background variation. Zelakiewicz et al. [196] developed a similar stand-off radiation imaging system (SORIS) carried by a van (Figure 13b). The detection system is a coded-aperture gamma camera composed by an array of four gamma cameras (each has 7.62 cm thick NaI plate with 60 cm × 46 cm) and a PMT readout (96 tubes). In this case, an active mask was used (23 NaI rods with 7.62 cm diameter) with the aim not only to attenuate the incident radiation—masking for localization purposes—but also to identify the radioisotopes—by spectroscopy. Some important characteristics to pinpoint are that it is a modular system that can be installed on air/sea platforms, uses GPS/inertial navigation system (INS) (increase precision and robustness), and has van side panels made with fiberglass to reduce gamma-ray attenuation. It also features an energy resolution of 8% at 662 keV. This detection system was developed to detect weak sources at large distances within a short time, which is especially important for search applications as is the case of homeland security. For example, an mCi source of ^137^Cs was located at 100 m stand-off using a 20–25 s integration time. Unlike standard coded-apertures, which have normally low efficiency (passive mask absorbs 50% of the incident gamma-rays) and FOV, a SORIS has the advantage of using a curved (enhanced FOV) and active mask (greater efficiency). Some limitations were reported, particularly the significant attenuation of lower gamma-ray energies by the vehicle cab and engine area, which leads to a reduced azimuthal FOV (vehicle dependent).

Within the scope of the MOBISIC project [197], a combination of a CZT detector (20 g) and a lightweight GAMPIX gamma camera (a 256 × 256 pixels matrix with 1.41 × 1.41 cm^2^ active area and a 1 mm thick CdTe substrate) with a 70 g coded mask (Figure 13c) was integrated in a small multi-rotor (1 kg MTOW) to be used in indoor environments (e.g., underground train tunnel) and for event securing. While CZT allows the first detection of the hotspot (by the increase of count rates), the gamma camera gives the source localization. The former can be used when the UAV is in the air or when landed (for identification purposes), and the latter is only used on the ground due to the need of long acquisition times (300 s) and bandwidth to transmit data.

#### 4.3.3. Pinhole Gamma Camera

Ohno et al. [198] used a Pinhole gamma camera together with a 3D-LiDAR (to adjust the gamma camera focal length) housed inside a truck to obtain the radiation image of the buildings and debris in the vicinity of FDNPP. Since radiation fields may achieve values of tens of mSv/h it was necessary to develop a heavily shielded operation box so that two operators can work safely. In order see the targets, a thermal camera and a pan–tilt–zoom camera are used. A gamma-ray sensor is also coupled to a remote-controlled robot (wireless communication) to measure the radiation field around the truck. A gamma dose (rate) meter with a telescopic probe (Teletector 6112D [199]) is also available when the workers need to leave the truck.

### 4.4. Combination of Neutron and Gamma Detection Systems

According to Section 2.2, the use of neutron detection together with gamma spectrometry is often considered and used for security-related scenarios, as the illicit trafficking of SNM and radioactive sources and materials.

Neutron detection systems are normally composed by three components: (i) a material to convert neutrons into charged particles or gamma rays, (ii) a sensitive volume to convert the resultant charged particles or gamma rays into an electric signal or light, and (iii) the data acquisition system. Examples of converter materials and their absorption cross section for thermal neutron detection are: ^3^He (5300 b), ^6^Li (940 b), ^10^B (3800 b), and ^156^Gd (61000 b). These materials are normally used in proportional counters or scintillators. Semiconductor detectors are expensive and, therefore, usually used in neutron imaging or neutron spectrometry. Most detectors used in nuclear security feature a higher sensitivity to thermal neutrons; therefore, a moderator can be used to slow down (mainly by elastic scattering) fast neutrons to increase the likelihood of their detection [19].

The detection of fast neutrons can be done directly by measuring the energy released by the recoil particles (nuclei) due to the elastic scattering of neutrons. The scattering material normally has a significant hydrogen content for higher energy transfer due to neutron elastic scattering. Examples of fast detectors are: stilbene, organic liquid scintillators with good pulse shape discrimination (PSD), PSD plastic scintillator (cheaper but PSD is worse than stilbene), PSD liquid scintillator (relatively inexpensive), and gas scintillator (e.g., ^4^He). Unlike thermal neutrons, the detection of fast neutrons allows determination of the direction of the neutrons (source localization and neutron imaging) and the source type (e.g., AmBe, fission, shielded) [19].

In order to have compact, lightweight, and lower power consumption detectors, dual-mode sensors can also be used to detect simultaneously neutrons and gamma-rays. The discrimination of either type of radiation is normally made by means of PSD. Some examples of dual-mode detectors are: organic liquid scintillators (e.g., NE-213, EJ-309), Cs_2_LiYCl_6_:Ce^3+^ (CLYC) scintillator (energy resolution 4.5–5%) [19]. The detection of neutrons in CLYC sensor is based on the ^6^Li reaction. Typical gamma energy resolution is less than 5% at 662 keV. Other compact dual-mode scintillators are: NaI(Tl+Li) [101], Cs_2_LiLaCl_6_(Ce) (CLLC) with an energy resolution of 3.4% at 662 keV, and Cs_2_LiLaBr_6_(Ce) (CLLB) with an energy resolution of 2.9% at 662 keV. Despite the fact that CLLB has better energy resolution, it features a worse PSD capability compared to CLYC or CLLC. Both CLLC and CLLB crystals present self-activity due to the presence of La-138 [19].

A compact CLYC detector was integrated in a multi-rotor for radiation safety and security applications (Figure 14a) [200,201]. The UAV can be used for monitoring, mapping, and source localization purposes (e.g., using the maximum likelihood estimation algorithm). Barzilov and Kazemeini [201] used the robot operating system (ROS) to implement a “plug and fly” concept and perform the data analysis and fusion functions on board. These data are timestamped and georeferenced to be sent to the user in real time.

Since security scenarios are normally characterized by weak sources or shielded sources, ground-based platforms, such as cars or trucks, are essential to carry large detection systems (∼1 m^2^) [202]. However, larger detectors also measure more background events and are more sensitive to background fluctuations [202,203].

Bandstra et al. [202,204] described the radiological multi-sensor analysis platform (Rad_MAP) system carried by a vehicle (Figure 14b) and composed by three types of radiation detectors: A gamma-ray imager with 10 × 10 array of NaI(Tl) detectors with a coded lead mask, an array of 14–24 HPGe detectors (for isotope identification), and an array of 16 liquid scintillators (EJ-209) for fast neutron detection. The Rad_MAP goal is to study variations in the natural background radiation and to serve as a development platform to mobile detection systems and imaging. To achieve this goal, it uses the information from several contextual sensors, like a panoramic video, a LiDAR (for data fusion of 3D scene reconstruction with gamma-ray images), a weather station, and a hyperspectral sensor, to analyze the impact of certain environmental parameters in the background. For example, it is known that the neutron background is mainly cosmogenic, and so it is lower and less variable than gamma-ray background. However, if weak sources of SNM are present, even small systematic variations in the background can influence its detection. The main contributor for the fast neutron background variability is the atmospheric pressure (affects the development and attenuation of the cosmic ray showers), followed by the “sky-view factor” (fraction of the sky that is not blocked by buildings), which is important in urban areas since there are more buildings and consequently more attenuation of the cosmogenic neutrons (lower neutron background). The authors concluded that there is a strong correlation between the surrounding environment and the background radiation and that the use of context and/or environment specific detection thresholds and background models could allow for improved detection sensitivity at a constant false alarm rate.

Within the scope of the project SLIMPORT, a sistema mobile per analisi non distruttive e radiometriche (SMANDRA, in English, stands for mobile system for non-destructive radiometric analysis) is being developed [205]. This mobile inspection system is divided into a passive unit for gamma-rays and thermal and fast neutrons detection and identification of radioactive material and SNM and an active unit composed by a sealed neutron generator to produce tagged neutron beams (the first unit is used to detect resulting radiation originated from the active unit). The photon spectrometry is accomplished by NaI(Tl) and LaBr_3_(Ce) detectors. The latter, despite the good resolution, due to internal activity, has some limitations in the detection and identification of weak ^40^K sources. Moreover, the volumes available are still limited. Therefore, a large NaI(Tl) is necessary to detect energetic gammas (6 MeV) originated by active techniques (inelastic excitation of oxygen and carbon). For weak and un-moderated sources, there is the interest of detecting fast neutrons since the STN ratio is higher compared to thermal neutrons, due to the dependence of 1/E of the cosmic-ray induced background. Therefore, an organic liquid scintillator NE-213 is used for fast neutrons and a ^3^He proportional counter for thermal neutrons. First results are in line with other studies [5], showing that small plutonium samples can be detected by using passive techniques. On the other hand, uranium samples are much more difficult to detect by passive techniques due to their low neutron yield and the ease of shielding gamma-rays (e.g., heavy metals); therefore, active techniques must be used. In a later work, Cester et al. [206] developed a compact gamma/neutron detector using the liquid scintillator EJ-309 coupled to a flat panel PMT. This detector has the advantage of lower toxicity and higher flash point compared to NE-213, making it more suitable for portable applications.

Cester et al. [207] developed a specific modular detection system, carried by a van, for the detection of SNM and other radioactive sources. It consists of a set of detectors based on high-pressure cells using noble gas scintillators filled with ^4^He for fast neutrons (better STN than ^3^He proportional counters with polyethylene moderator) and Xe for gamma-rays. It was also developed ^6^Li-lined ^4^He tubes, allowing to detect (using PSD techniques) either fast and thermal neutrons. A set of eight fast neutron detectors (FND) and two ^6^Li-lined thermal neutron detectors (TND) were used. Xe-based scintillator was chosen mainly because of its robustness, vibration insensitivity, non-hygroscopic properties, volumes available, and better energy resolution than NaI (6.7% at 662 keV). Additionally, there was a NaI(Tl) scintillator (125 × 125 × 250 mm^3^) to maximize the detection efficiency at energies above 1 MeV. The detection system is capable of detecting and identifying gamma sources (NORM included) and neutron sources, like ^252^Cf, Am/Be, Pu/Be, and SNM, as well as the presence of hydrogen-rich or lead shielding surrounding neutron sources. The detection system was tested according to ANSI 42.43, complemented with field tests using moving sources ("drive by" mode) and maritime cargo containers. The field tests allowed to pinpoint the importance of natural background variations which had an impact on the measurements (e.g., occasional triggering of false alarms).

The mobile urban radiation search (MURS) project [208] consists of a real-time detection, identification and localization of radiological and nuclear sources in urban environments. The detection system is mounted in a car (Figure 15) and is composed by a six 5.08 × 10.16 × 40.64 cm^3^ NaI(Tl) scintillators and a ^6^Li neutron sensitivity layer. Additional contextual sensors, like visible and near-infrared cameras, were used to improve scene awareness. A GPS stabilized by an INS helped to have position information in case of GPS failure. Other features presented were temperature control and radionuclide list mode.

For the passive neutron detection and localization of SNM, Hutcheson et al. [209] proposed a containerized detection system carried by a truck (Figure 16a) and composed by two separate subsystems: i) for TND, 24 BF3 (diam. 11.4 × 183 cm, 93.2 kPa) and six ^3^He detectors (diam. 14.7 × 64 cm, 279 kPa) were considered, both surrounded by 2.5 cm high dense polyethylene (HDPE), and, ii) for FND, a 6 × 8 array of EJ-309 liquid scintillator detectors (15.2 × 15.2 × 15.2 cm^3^) coupled to PMT. HDPE shielding was placed around the array except on one side for electronics access. Other features included a pair of GPS, a temperature and humidity control, a diesel generator to guarantee autonomy for 5 days, and a UPS to protect from power fluctuations. Static and in motion measurements were performed using a ^252^Cf source (775,000 neutrons/s). Some results obtained were: (i) for stationary detection the TND would give an alarm (successful detection) with a 1 s integration time at 60 m and with a 5 s at 100 m; (ii) with a vehicle speed of 2.7 m/s, it was possible to detect the source at standoff distances of almost 100 m and 70 m using the FND and TND systems, respectively; and iii) the background count rate is lower 20% in water than in land—due to the hydrogen content in water—allowing neutron alarm at greater distances for on-water missions. The authors concluded that it would be extremely useful to localize neutron sources, which, for fast neutrons, could be accomplished with the addition of a HDPE coded mask, while, for thermal neutron, it would have to be achieved with another approach.

The real-time wide area radiation surveillance system (REWARD) project [210,211,212,213] consists of the integration of a gamma spectrometer and a neutron detector based on recent silicon technology for, inter alia, homeland security. Despite the security purpose of this project, it can also be useful in scenarios as loss of a radioactive source and radioactive contamination due to nuclear accident. The main goal is to have various units composed by the detection system, GPS, and a wireless communication interface located in infrastructures (e.g., buildings) or carried by vehicles in order to send data to a monitoring base station, which will emit an alarm in case of source detection. The gamma spectrometer consist in two stacked 1 cm^3^ CZT detectors working in coincidence. This configuration allowed increasing of the total efficiency of the system (without increasing to much the cost) and reduction of the Compton background. The neutron detection system comprises a slow and a fast component detectors (Figure 16b). The slow neutrons are detected by using micromachined silicon structures backfilled with a boron converter, while the fast neutron component is detected with thin planar silicon p-in-n diodes covered with a layer of hydrogenated plastic radiators. For neutron moderation, a polyethylene cube is also used, and all sensors are surrounding it.

Kazemeini et al. [214,215] implemented a plug-and-play concept for the integration of radiation sensors into robotic platforms (multi-rotor and a ground vehicle). The detection system consists of the combination of a high-resolution gamma spectrometer CZT and a CLYC scintillation crystal with a Lithium-6 enrichment of 95% for both gamma and neutron detection (using PSD technique). In this work, the use of ROS to integrate sensors onto the platforms and for data fusion (time and position for each radiation measurement) was considered. These data will allow a cooperative radiation monitoring using multiple UAVs and UGVs (swarms) for radiation source localization and contour mapping.

In order to improve the detection probability of SNM materials, Sullivan et al. [216] proposed a network of low-cost mobile detectors (e.g., PRD Kromek D3S), which, in this case, are carried by persons to measure gamma and neutron radiation. Despite the use of data fusion techniques (e.g., geospatial alignment by universal Kriging method) and a Kalman filter to avoid discontinuous or missing values from the GPS signals, differences in the average count rate were reported (almost 40%), so a different approach will be necessary in the alignment of the sensor measurements as the use of machine learning techniques for an adaptive sensor normalization. The continuous data acquisition can also benefit the study of the background fluctuations in time and space due to weather conditions.

Due to the worldwide shortage of ^3^He gas, it is crucial to find an alternative detection technology for homeland security. Although Kouzes et al.’s [217] study is related to RPMs, it is worth mentioning that, among several detectors analyzed (BF3, boron lithium-loaded glass detector, and coated plastic fiber detector), the best candidate to replace ^3^He technology is the boron-lined detector since it has good neutron detection efficiency; gamma ray insensitivity; unlike BF3, is not a hazardous material; and has a reasonable operating voltage.

With the same goal, a European project called SCINTILLA [218,219] developed and tested some ^3^He alternatives for the detection of nuclear sources using RPMs: the EJ200 plastic scintillator, Gd-lined plastic scintillator, and a LiZnS neutron sensor. For the test and benchmark of detection systems, it was used a facility which can reproduce many scenarios as moving sources (at fixed portals) or search with moving detectors. This international platform allows testing of the response of different detection systems to several sources available: neutron and gamma response (e.g., using different shielding), the influence of gamma radiation on neutron response, the radionuclide categorization/identification (SNM, NORM, medical and industrial isotopes), and by using different masking scenarios (e.g., SNM with NORM or medical isotopes). SCINTILLA also considered spectrometric personal radiation monitor technologies, like the scintillators and CZT detectors (e.g., CZT gamma cameras), for the radionuclide identification and masking scenarios.

### 4.5. Dual Particle Imaging Systems

In 2011, Polack et al. [220] suggested the concept of dual-particle imaging system by combining Compton scattering and neutron scattering techniques to detect fast neutrons and gamma rays. A simultaneous gamma and neutron imager has many advantages relative to a single particle imager described in Section 4.3, since it allows the detection of more nuclear materials and other radioactive sources, making it more difficult to shield radiation from concealed sources (by detecting fast neutrons) and improving performance in areas with high levels of gamma background. The dual particle imaging can be divided into two groups: (i) imager composed by detection materials that are both sensitive to gamma-rays and neutrons; and (ii) imager composed by various detection materials which might not be sensitive to both types of particles. Normally, the latter has some advantages, such as less complex systems, no particle discrimination techniques, and higher design flexibility [21].

Al Hamrashdi et al. [221] recently developed a portable dual-particle imager for the combined detection and localization of gamma-rays, thermal, and fast neutron sources for applications, such as nuclear materials assay and nuclear non-proliferation. The design and materials optimization were reported in References [222,223], which led to a three-layer configuration composed by the scintillating materials: GS10 lithium glass, EJ-204 plastic scintillator, and CsI(Tl) inorganic scintillator. Despite the promising results (using a scan time of 60 s), future work is still needed to improve the discrimination between the neutron and gamma pulses.

Recent research also identifies scintillator-based coded-aperture imaging systems to be used in real-time and portable equipment for neutron localization as emerging promising solutions in security and nuclear decommissioning applications [20,224]. However, there is still a need to improve the coded-mask techniques and the neutron detection sensitivity across the energy spectrum between thermal and fast neutrons (up to 15 MeV). New techniques in crystal growing allowed to consider the single stilbene crystal (solid organic scintillator) as a good alternative to the widely used liquid scintillator technology (e.g., EJ-309), since the former is not susceptible to leaks, is less hazardous, and claims better PSD and energy resolution, as well as increased neutron efficiency, due to the very low threshold (∼20 keV), while maintaining a high gamma/neutron discrimination [225]. Moreover, stilbene crystal have better gamma/neutron discrimination performance than plastic scintillators. However, stilbene is more expensive (machining costs) and is more fragile, leading plastic scintillators, such as EJ-276 or EJ-276G (formely EJ-299-33A, EJ-299-34) from Eljen Technology, as a good alternative for coded-aperture imager usage [224,226]. New coded-aperture masks are also necessary to have compact and lightweight equipment for portable applications.

Finally, Ayaz-Maierhafer et al. [227] studied the effects of the angular resolution of a combined gamma-neutron coded-aperture imager for the standoff detection of a nuclear threat source (e.g., orphan source), and Soundara-Pandian et al. [228] proposed a portable gamma-ray/neutron imaging system consisting in a CLYC scintillator with a coded-aperture mask of Cadmium and Tungsten for security applications.

### 4.6. Challenges and Research

The need for portable detection systems for real-time radiation detection of radioactive and nuclear materials entails radiological and technological challenges. The most challenging issues are related to the small physical size of the radiation sensors and the large source-detector distances (particularly for scenario B), which impacts the counting statistics and accuracy of the results [212]. Other parameters, such as the energy resolution of the detectors (for identification purposes), the time required for operation, environment conditions, etc., must also be considered.

Mobile radiation sensors should be lightweight and low power so that they can be integrated into the robotic platform without affecting data processing, payload, or battery power [214].

The main challenges and research encompass:Integration of contextual sensors based on ground-penetrating RADARs (GPR) in radiation imagers [229]—normally, the contextual sensors used to characterize the distribution of radiation sources are based on visual sensors. However, in decommissioning tasks, it is necessary to check the origin of the radiation deep inside materials, as is the case of contaminated pipelines which are underground or inside concrete, or the ingress of radioactive contaminants in concrete. Therefore, to improve the 3D localization of the radiation source in depth, it is important to develop 3D reconstruction algorithms based on the fused data of a gamma imager and a GPR.Environmental factors—weather conditions may be difficult and influence the radiation measurements. Urban areas are complex radiation environments (e.g., different structure materials) and pose many challenges in terms of vehicles access, shielding, and potential for concealment of sources, communications, etc. [9,126].Use of gamma cameras coupled to small UAVs [187]—these platforms are extremely maneuverable and can be used for autonomous source localization; however, some improvements are necessary in the development of compact and lightweight gamma cameras, reduction of the acquisition times, and image compensation due to the movement of the source or detection platform, while acquiring the gamma image [230].Detection system change—normally, radiation detection systems are expensive, and there is a reluctance in changing them by new ones (using emerging technologies). In some cases, both types of equipment (the old and the new one) are used in parallel. Therefore, in order to keep using the old equipment, its important to consider the data transmission methods, data formats, and analysis algorithms [12].Remote expert support (reach-back)—the remote analysis capability allows each detection instrument not to include all the functionalities. The information may be sent to a remote server that, in turn, is analyzed by an expert [12].Network of detection systems—the increasing requirements of data storage or data transfer between detection units and a reach-back center (e.g., for secondary analysis) may be obstructed by cyber-attack or equipment/connection lost. Therefore, a fast, reliable, and secure network connection with sufficient bandwidth is necessary [9,12].Interoperability between detection systems—the data formats must be standardized and, at the same time, flexible for sending diversified information. For homeland security scenarios, there are specific standards for data formats (see Section 2.2) [9,12].Activity estimation of radioactive sources—this can be challenging for mobile detection systems since, usually, the geometry is unknown; for example, there is no a priori knowledge of the shielding material between the detector and the source [12].List-mode data acquisition—this feature consists of recording the output data of a given detector. Timestamped list-mode allows the automatic comparison of the outputs of different detectors, for example, to reject false alarms and for source localization. However, difficulties may appear when synchronization of mobile systems is necessary with high accuracy (e.g., fast detectors) [12].Alternatives to Helium-3 neutron detectors due to the worldwide shortage [217,219].Use of multiple UAVs for the detection and localization of source(s), for example, by using plastic scintillators (poor energy resolution but cheaper)—the source identification could be done afterwards using a NaI system or other inorganic scintillation detector [89]. Other studies suggest the use of energy windowing to distinguish SNM from NORM [231] or by using deconvolution methods of the spectrum acquired by plastic scintillators [12]. These scintillators can also be loaded with heavy elements (e.g., bismuth) to increase the photopeak efficiency; however, this decrease the light yield [232].Detection materials improvement for mobile applications—research is necessary to improve the performance of the detection systems available and find new ones. An example is the fast-growing development of halide perovskite radiation detectors (e.g., halide lead perovskite). These semiconductors present some advantages as their low cost, room temperature operation, high stopping power, defect-tolerance, and high energy resolution compared to NaI(Tl) scintillator, which makes then a promising candidate in X-ray imaging and gamma-ray spectroscopy. Research is still necessary to improve their characteristics, e.g., sensitivity, dark current density, resistivity, and environmental stability (to heat and moisture) [233].

Other challenges are related to radiation hardened materials, as well as the effects of magnetic field on the detection systems.

According to IAEA [27], the areas related to detection technology that should be considered for future research are: Detection probability and range, source identification, and mobility.

## 5. Results and Discussion

From the analysis of the four scenario types considered here, Table 6 was constructed, which summarizes their main characteristics and operational restrictions (difficulties) and the corresponding desirable characteristics of the mobile radiation detection systems. Some general characteristics of a mobile detection system are also presented.

For a given scenario, there are the following radiation measurement options: localization, identification, and quantification (of the activity) of a hotspot or contaminated area or just obtaining a contamination map.

Radiation detection systems can be coupled to many mobile platforms. To choose the best platform to perform a measurement, one must know the advantages and limitations of each platform:Ground-based platforms, as cars, vans or trucks are normally chosen due to their greater payload capacity and autonomy. However, they can only move through the existent road network, which may present obstacles (e.g., post-disaster scenario). Handheld or backpack equipment allow access to difficult places and to have an excellent spatial resolution. The UGVs allow operation in extreme situations where radiological risks are unknown or too high to be done by humans. Issues, such as communication problems, radiation tolerance, and obstacles and terrain limitations, were reported.Underwater platforms are currently used to explore inactive mines (flooded tunnels or lacks) to obtain topographic, geological, and mineralogical information. Since these sites have many metallic and industrial products, their use in the future to assess in situ the radiation levels in samples must be equacionated. Small UUVs are used in the visual inspection and non-destructive evaluation of an NPP (e.g., pressure vessels or other water-filed infrastructures).Air-based platforms. Manned fixed-wing aircrafts allow very large area coverage, great payload capability, faster deployments, and surveys. Manned helicopters have the advantage of VTOL and loitering features, i.e., can substantially reduce their speed or even hover on top of a given area. However, both manned air-based platforms are limited to a minimum flight altitude (safety altitude), below which they cannot fly. Moreover, when radiation levels are too high, the radiological risk to the crews may not allow the measurements. Unmanned fixed-wing aircrafts can also be used to cover large areas and perform fast deployments without putting humans at risk. Despite the fact that unmanned helicopters have lower speed compared to fixed-wing counterpart, they allow VTOL and loitering, which allows lower altitude measurements (higher spatial resolution). Multi-rotors also feature VTOL and loitering capability, allow greater spatial resolution (can fly at 1 m AGL if necessary), and are easier to operate. However, it is the most limited platform in terms of payload and autonomy.

Unmanned systems can be used as autonomous platforms for the search of hotspots (radioactive sources) or the mapping of contaminated areas. They can also be used as swarms to overcome obstacles, autonomy issues, and optimize the search of hotspots. Cooperation between unmanned platforms was also referred in literature, as the use of a UAV to collect terrain information (using photogrammetry) to help a UGV (in the path planning) for the search of a hotspot in an unknown environment.

Special challenging environments for unmanned vehicles, particularly UAVs, are urban areas and indoor environments. These environments are characterized by navigation (GPS failure) and communication issues (losing connection or control signal latency), possible collisions with infrastructures/persons or other low altitude aircrafts (outdoor), and noise.

Despite the advantages of unmanned blimps (e.g., long endurance for monitoring tasks), hybrid platforms, such as VTOL fixed-wing (no runway needed; however, it is a complex system) or PLIMP (blimp-plane aircraft), as far as the authors know, have not been used for radiation monitoring for the scenarios considered here.

An important area of research is the development of bio-inspired robots for use in disaster scenarios due to their adaptability and fault tolerance, using, for example, the modular design concept. Some bio-inspired robots are already commercially available and can be used in disaster scenarios. However, it is still necessary to study the radiation hardness of the different electronic components and sensors.

Table 7 summarizes the mobile detection systems and platforms described in literature for the four scenarios considered in Section 2 and their advantages and limitations. Some considerations are discussed below.

Due to their good intrinsic efficiency, reasonable energy resolution (spectroscopy properties), and availability of sensitive volumes of reasonable dimensions, inorganic scintillation detectors are used in all scenarios. From the literature review, the most frequently used gamma scintillation radiation detection and spectrometers used in mobile platforms are: BGO, CsI(Tl), CsI(Na), NaI(Tl), LaBr_3_(Ce), and GAGG(Ce). A widely used alternative is the semiconductor CZT, which works at room temperature and has better energy resolution; however, it is limited to small crystals (a few cm^3^). There are scenarios where specific scintillation crystals have advantages relative to others due to their efficiency, energy resolution, and time response characteristics. Moreover, the necessary robustness, weight, and size of the detector system may limit its use in certain platforms. Namely:LaBr_3_ is essentially used in the scenarios A and C (high dose rates scenarios) because of its fast response, good photoelectron yield (compared to the traditional NaI), and good energy resolution. For scenario B and D (low dose rates), the use of this detector is limited due to the crystal self activity (La-138), which can superimpose the natural background counts (particularly the NORM K-40).BGO detectors can be used in situations where good counting efficiency and mechanical characteristics are necessary, in detriment of energy resolution.The availability of larger area SiPM, and the fact that these photosensors are smaller, lower power, and lighter than the traditional PMT-based scintillation detectors, made possible the use of CsI(Tl) and LaBr_3_ scintillation crystals in small unmanned vehicles and as Personal gamma spectrometer, PRD, or handheld equipment. The Ce:GAGG scintillator coupled to SiPM was also used as a Personal gamma spectrometer.CZT detectors were used in scenarios A, B, and D. Despite the fact that they can be used in small UAVs (multi-rotors), due to efficiency limitation related to the small crystal volumes available, it becomes necessary to fly at very low altitudes (1–15 m) and speed (<1.5 m/s) in order to detect hotspots or contaminated areas in low dose rates scenarios (B and D). This may be impracticable due to ground obstacles (e.g., tall vegetation).CsI(Tl) scintillators are used in scenarios A and B due to their reasonable resolution and good mechanical and chemical properties (slightly hygroscopic). It is a light-weight scintillator solution.HPGe semiconductors are heavy and high energy consumption detection system (need cooling system); however, due to their excellent energy resolution and high efficiency, they are used in mobile platforms with greater payload capacity as a van (with or without shield) or a manned airplane. They were also used as a backpack system; however, its 25 kg weight is a limiting factor.

Gamma imagers (summarized in Table 8) are normally used in scenario A, for example, to source location (hotspots), to evaluate decontamination efforts, and for assessment of the contamination distribution and mobility (e.g., decommissioning of FDNPP). Despite the difficulties in using gamma cameras in fast detection of weak sources (e.g., SNM), their use is also referred in scenario B, particularly the larger volume Compton camera SCoTSS and the use of 3D Compton imaging (VCI), for source searching and mapping of unknown environments.

Although there are three types of gamma cameras, the most common used gamma cameras in mobile systems are the Compton cameras and the coded-aperture cameras.

Compton camera. Since Compton cameras do not use collimator, they are lighter and can be used in unmanned helicopters, multi-rotors, crawler robot, or even as handheld equipment. For these mobile platforms, the scatter and absorber scintillation crystals used were the Ce:GAGG coupled to SiPM and APDs or to MPPC, which allowed compact and lightweight gamma cameras. The choice of Ce:GAGG crystals is related to its low cost (compared to NaI and LaBr_3_:Ce), acceptable energy resolution, high light yield, and good Compton efficiency. The first portable Si/CdTe Compton camera was also developed, with very good energy resolution; however, this system weighs 10 kg and needs an exposure time of 30 min. Normally, the Compton gamma image is obtained when the platform is stopped or hovering; however, it is also possible to perform radiation imaging while the platform is moving by knowing its position and posture of the camera for each detection (extracting the Compton cones direction). Since the gamma image reconstruction can take minutes and since multi-rotors have very low autonomy (16 min), the importance of developing a 4π Compton camera to obtain the contamination image inside a building is highlighted. For the source searching and mapping on unknown environments or to localize sources inside objects, a 3D Compton imaging system was also developed, which used scene data fusion of the gamma image with a 3D scene reconstruction using a Kinect sensor, a photogrammetry software, or a 3D-LiDAR. SLAM techniques are used to obtain the position and posture of the imaging system. In addition to the use of Ce:GAGG crystals for VCI, the use of HPGe detectors and a high-efficiency multimode imager, which allows both coded-aperture and Compton imaging modes, is also reported and described. Despite the fact that VCI are being implemented in handheld equipment and in mobile platforms, improvements are still necessary in the 3D scene data fusion, an important tool not only for detection and mapping gamma sources but also other particles (e.g., neutrons).Coded-aperture camera. To detect weak sources at larger distances, as is the case of scenario B, larger gamma cameras were developed to be carried in a cargo trailer (SORDS-3D) or by a van (SORIS). While the former used coded-aperture camera composed by CsI(Na) scintillation crystals and a passive mask, the latter used NaI crystals and an active mask used not only to attenuate the incident gamma-ray but also for spectroscopy purpose (increased efficiency). The use of a lightweight coded-aperture gamma camera (GAMPIX) coupled to a multi-rotor is also referred to in the literature.Pinhole camera. Since pinhole cameras are heavy and low efficiency equipment (thick collimation system), they have few applicability to mobile platforms.

The use of gamma cameras for autonomous mapping and hotspot localization using mobile platforms is still limited by the long acquisition times, poor angular resolution, difficulty in detecting distant hotspots when intense sources are nearby the camera or when considering measurements with a moving source/camera (blurred images). Another challenge is related to the determination of the activity of a given hotspot from a gamma image (and depth information).

The detection of SNM materials and other radioactive materials in nuclear security applications is very difficult (scenario B). The SNM are normally weak gamma sources and can be shielded or masked (e.g., NORM radionuclides). Moreover, some Pu and U radioisotopes decay by spontaneous fission, emitting neutrons that can be used to detect these materials. Therefore, it is normally used a combination of neutron (passive or active detection) and gamma-ray detection system in this scenario. In order to have confidence in these detection systems and to not limit the people and vehicles traffic (inspection sites as entry points), standards are available to define requirements, such as sensitivity, discrimination capability (gamma detectors must be insensitive to neutrons, and vice versa), and admissible rate of false alarms. To achieve these requirements, normally, multiple detectors with very large volumes (high efficiencies) are used, carried by a van or a truck. The detection system must be also capable to detect neutron sources in the presence of hydrogen-rich or lead shielding. Detectors used for thermal neutrons are normally BF_3_, ^3^He tubes surrounded by HDPE or ^6^Li-lined detectors, while, for fast neutrons, liquid scintillator detectors, like EJ-309 or gas scintillators filled with ^4^He, can be used.

Compact detection systems to measure both gamma-rays and neutrons can be achieved, for example, with the use a dual-mode CLYC scintillation crystal with a ^6^Li enrichment (95%) using PSD techniques. An alternative to large detection systems is the use of compact detectors coupled to multiple UAVs and UGVs for cooperative source localization and contour mapping or the use of a network of low cost mobile detectors to measure gamma and neutron radiation by using data fusion techniques.

In the last decade, dual-particle cameras began to be developed and deployed to account for both Compton scattering of gamma-rays and neutron scattering. Compared to single particle imagers, these cameras have better performance in high gamma backgrounds and allow to detect more nuclear and radioactive sources, particularly shielded sources (by detecting fast neutrons). Work is still needed in the discrimination of neutron and gamma signals.

Recent research considers the use of scintillator-based coded-aperture imaging systems for real-time neutron localization in portable equipment for security and nuclear decommissioning applications. Some of the scintillator candidates are the stilbene crystal (more expensive and fragile) or the plastic scintillator EJ-276 from Eljen Technology.

Finally, contextual sensors are essential for the description of the environment as 3D scene reconstruction (e.g., Kinetic sensor or 3D-LiDAR), as well as in the analysis of the influence of environmental parameters (e.g., by using a meteorological station) in gamma and neutron background. Notwithstanding the fact that all contextual sensors are restricted to obtain visual information of the surface, there is an increasing interest in using ground penetrating RADAR together with gamma cameras to obtain contamination distributions in depth, as in the decommissioning of a nuclear power plant (e.g., inner pipelines or depth contamination in concrete).

## 6. Conclusions

During the last decade, new developments in mobile platforms, radiation detection systems, and contextual sensors brought new challenges to the detection, localization, quantification, and identification of radionuclides, as well as on the mapping of contaminated areas. An example was the first use of small UAVs to carry lightweight and compact radiation detection systems in the aftermath of the Fukushima NPP accident (2011).

Since then, various combinations of sensors and platforms were suggested and implemented to achieve the best results for a particular scenario. Some scenarios pose significant technological challenges, addressing multidisciplinary issues to, inter alia, radiation detection, instrumentation, sensors, electronics, information and communication technologies, and robotics.

This review aimed at providing a broad perspective of the recent developments in mobile radiation detection systems, particularly for gamma-rays and neutrons, referring the advantages and limitations of each radiation detection system and platform used, considering four different scenario types:Scenario A: is related to RN accidents and emergencies due to incidental or intentional release of radioactivity in the environment, as is the case of a nuclear power fission reactor accident or in the event of a malicious act, respectively. Includes emergency decommissioning (post-accident) of nuclear facilities and long-term monitoring. In this scenario, one might be interested in detecting, localize, quantify, and identify the source(s) or just map the radionuclide distribution (e.g., comparison over time of the effectiveness of remediation processes for a given area).Scenario B: the goal is to detect, localize, and identify SNM and radioactive materials as a consequence of illicit trafficking or inadvertent movement. In this scenario, the detector-source distance can be considerable (up to 100 m), and the SNM are weak gamma sources that can be easily shielded or masked (medical isotopes and NORMs) must be considered. Both gamma and neutron detection systems (some SNM radioisotopes are neutron emitters—spontaneous fission) are normally considered for this scenario. Active techniques may also be needed to detect SNM, particularly highly enriched uranium.Scenario C: related to the detection and localization of leaks of radioactive materials and quantification of the levels of radioactivity (e.g., generated by activation products), when performing the inspection, maintenance or repair of nuclear, accelerator, targets, and irradiation facilities. It also includes normal decommissioning and long-term monitoring.Scenario D: involves the detection, localization, monitoring, and quantification of NORM concentrations and their identification. Since NORM distribution may vary due to variation of mineral content in soils or due to human activities (e.g., ore extraction), it is generally necessary to map a region of interest and monitor the radioactivity levels.

The scenario type (e.g., characteristic dose rate, radionuclides to be detected), the survey area, or whether it is an emergency situation or a normal measurement (fast or slow survey), will impact and constrain the mobile platform and detection system chosen.

If large and heavy detection systems are necessary (to increase the efficiency), manned aircrafts and vehicles are normally used. Manned aircrafts have the advantages of performing a fast deployment and survey (large area coverage); however, the high minimum altitude (approximately 150 m) and typical speeds limits the spatial resolution of the measurements. Manned vehicles can only travel in the road network, and their movement can be quite limited if there are obstacles on the ground, such as in an emergency (e.g., Fukushima accident). Foot-based surveys, normally made with the help of PRDs, handheld, and backpack equipment, can obtain excellent spatial resolution but are limited by the dose limits imposed to humans.

When the radiation field is unknown or the associated doses may be too high for humans, a good alternative to manned vehicles or foot-based surveys are the unmanned systems. UGVs can be used to carry heavy payloads; however, they have several issues related to communication problems (e.g., inside buildings) and are limited by obstacles and terrain. UAVs can overcome obstacles on the ground, can be deployed and fly faster to the area, and perform larger surveys at lower altitudes and at lower speeds (improved spatial resolution compared to manned aircrafts). Due to their higher maneuverability, multi-rotors (aka drones) are normally the preferred option to perform measurements in urban areas or indoor environments.

To optimize the use of unmanned systems, some challenges still need to be overcome, namely: the development of specific mobile robots for RN scenarios, autonomous operation, robot learning, battery autonomy, affordable, faster, and more efficient sensors.

Due to their spectrometric properties and high efficiency, inorganic scintillation detection systems are used in all scenario types for detecting, identifying, and/or searching a hotspot, or just mapping a contaminated area.

LaBr_3_(Ce) scintillator has become the new standard due to its good energy resolution (2.6 % at 662 keV) and fast response (16 ns). The LaBr_3_ (Ce+Sr) scintillating crystal with enhanced properties (energy resolution of 2.2% at 662 keV) became recently available; however, it has to be tested in the field. Both detectors are appropriated for scenarios A and C. The intrinsic radioactivity of these crystals (due to the presence of ^138^La) makes it difficult to use on scenarios B and C (characterized by low dose rate). An alternative may be the use of CeBr_3_ scintillator crystals.

Scintillator crystals, such as Ce:GAGG, CsI(Tl), and LaBr_3_, can be coupled to SiPM photosensors, making them lighter, small, and less vulnerable to magnetic fields (unlike PMT coupling). However, the use of SiPM on scintillators may result in lower energy resolution and greater temperature dependence. The use of lightweight SiPM-based scintillators allows their integration in small mobile platforms, as small unmanned systems, or in backpack or handheld equipment. Some other SiPM-based scintillators are becoming commercially available (Saint Gobain), such as the the NaI(Tl), NaIL (dual mode detector for gamma radiation and neutron particles), LaBr_3_:Ce, and enhanced LaBr_3_:Ce crystals.

The most versatile detection systems, used in scenarios A, B, and D, are the NaI(Tl) scintillator detector and the CZT semiconductor. The NaI(Tl) scintillator is available in large crystals, has a lower cost, and features a reasonable energy resolution, while the CZT features good energy resolution (better than scintillator detectors) and is available in small crystals (a few cm^3^), which might limit its use for low dose scenarios (B and D).

Some lightweight and compact COTS detectors are available, such as the silicon-based technology (e.g., Raspix), CZT semiconductor, and SiPM-based CsI(Tl) scintillator.

An alternative to spectrometry techniques for the search of hotspots is the use of directional detectors, such as gamma cameras. Since Compton cameras are lighter, they can be coupled to small mobile platforms (e.g., UAV and handheld equipment). The Ce:GAGG scintillator crystals is the most popular material for the construction of such cameras due to their low price and possibility to be coupled to SiPM and APDs (lighter detectors). For bigger mobile platforms housed and transported in vans and trucks, coded-aperture cameras are used to detect weak sources at large distances (scenario B). Due to their heavy collimation system, pinhole cameras are not very common in mobile platforms. In the literature, gamma cameras can be used in scenarios A and B; the latter represents a greater challenge due to low dose rates.

For the detection of SNM (scenario B), it is normally necessary to consider the use of large volume detection systems (carried by vehicles), both for gamma-rays and neutrons. For thermal neutrons, normally BF3, ^3^He tubes surrounded by HDPE or ^6^Li-lined detectors are used, and, for fast neutrons, the most common is the EJ-309 liquid scintillator. An alternative is the use of lighter and compact detection systems, such as the dual-mode CLYC scintillation crystal with a ^6^Li enrichment (95%) (using PSD techniques). The latter system can be carried, for example, by the highly maneuverable multi-rotor.

Another promising compact detection system uses the phoswich (phosphor sandwich) detectors, in particular, for the measurement of low intensity uranium samples in a high energy background.

The recent development of dual-particle cameras, for measuring Compton and neutron scattering, are promising detection systems since they allow more SNM and radioactive material to be detected (scenario B), as well as for shielded and masked sources.

In the same way, scintillator-based coded-aperture imaging systems are being considered for real-time neutron localization in security and nuclear decommissioning applications. Some of the scintillator candidates are the stilbene crystal (expensive and fragile) or the plastic scintillator EJ-276 from Eljen Technology.

Several standards are available for mobile detection systems in homeland security (e.g., handheld and vehicle-based equipment); however, to the best of the authors knowledge, there are no standards for the detection systems coupled to air-based platforms. Similarly, there are no standards for gamma cameras.

Promising future research, technology, and innovation topics encompass: (i) Mobile platforms with improvement of unmanned vehicles, particularly small UAVs, for urban areas and indoor environments operation (e.g., GNSS signal transmission problems, obstacles), use of hybrid unmanned vehicles (e.g., VTOL fixed-wing, PLIMP airship), bio-inspired robots to overcome obstacles and terrain irregularities, battery autonomy improvements, swarm and cooperative robotics (e.g., cooperative navigation between UGV and UAV), and radiation tolerance of electronic devices; (ii) algorithms, data processing and transfer, implementing artificial intelligence, deep learning (robot learning and on-board processing) and computer vision algorithms (e.g., for the autonomous localization of a radiation source), and acquisition and treatment of real-time data, big data fusion, and detection system networking (fast, reliable, and secure network connection); and (iii) detection systems, including the interoperability between them (e.g., standardization), alternatives to helium-3 neutron detectors, improvement of existent detection systems and search for new detection materials with higher detection efficiency and energy resolution featuring significantly improved characteristics concerning compactness, robustness, reliability, low-cost, low power consumption detectors for use in small mobile platforms (e.g., unmanned vehicles/robots). Concerning new or improved detection systems, some important areas of research, technology, and innovation activities are related to the development of dual-mode detectors (e.g., neutron and gamma detection), phoswich detectors, halide perovskite detectors, SiPM-based scintillators and plastic scintillators, and compact and real-time gamma cameras and dual particle cameras (e.g., use of stilbene or plastic scintillators) for mobile platforms. Finally, understanding and mitigation of the effects of strong magnetic fields and other environmental factors (e.g., temperature) in the detectors’ behavior will presumably attract the attention of the communities of experts.

## Figures and Tables

**Figure 1 sensors-21-01051-f001:**
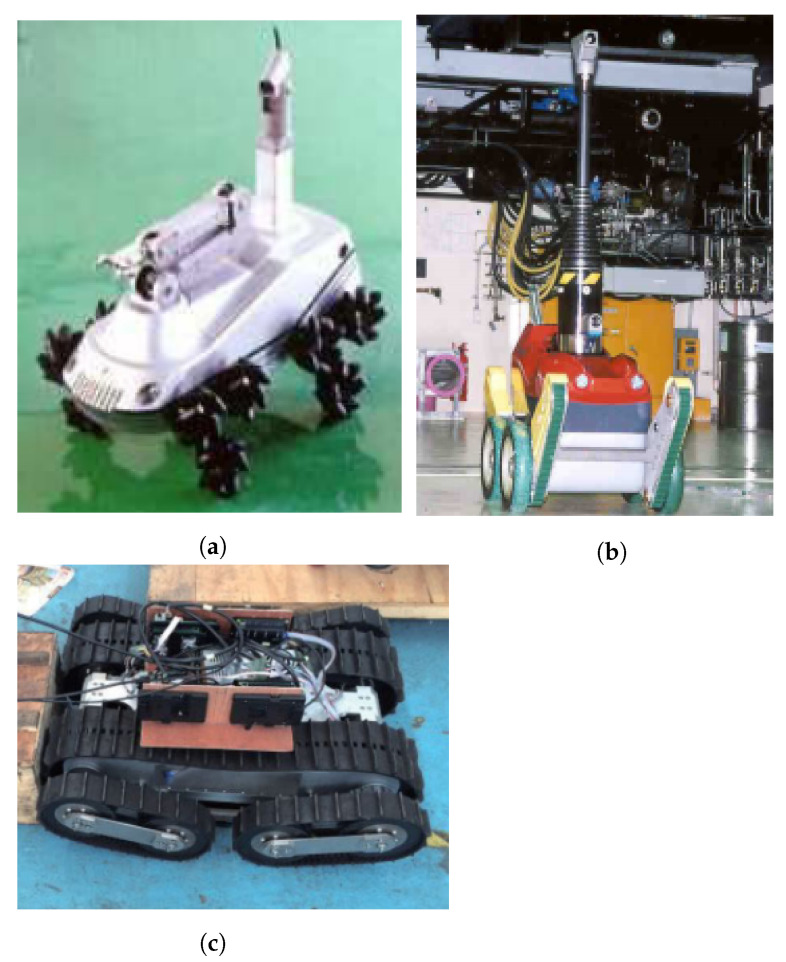
Examples of unmanned ground vehicles (UGVs) types: (**a**) UGV with wheels and a manipulator. ©2010 IEEE. Reprinted, with permission, from Reference [47]. (**b**) UGV with wheels and four tracks. ©2010 IEEE. Reprinted, with permission, from Reference [47]. (**c**) UGV with tracks. ©2014 IEEE. Reprinted, with permission, from Reference [46].

**Figure 2 sensors-21-01051-f002:**
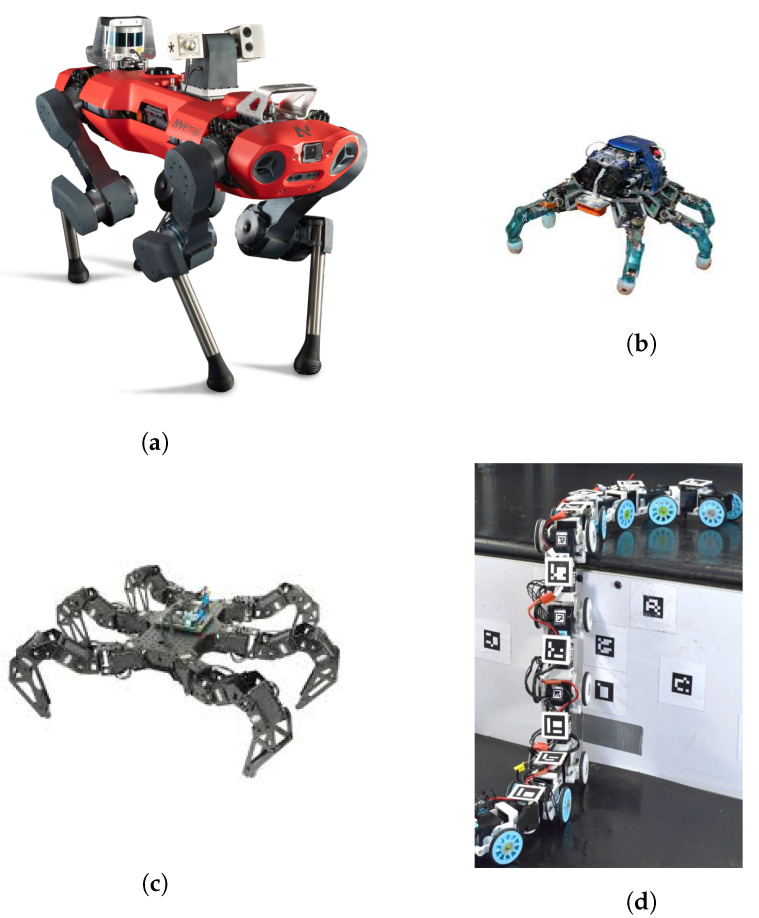
Examples of bio-inspired robots: (**a**) Anymal C from ANYbotics. Reproduced, with permission, from Reference [51] copyrighted 2021 ANYbotics https://www.anybotics.com (**b**) DLR (CC BY 3.0) crawler. Reproduced, with permission, from Reference [53]. (**c**) Hexapod MKIII. Reproduced, with permission, from Reference [55]. (**d**) Snake-like robot. Reproduced, with permission, from Reference [56]; published by the Royal Society, 2020, under the terms of the https://creativecommons.org/licenses/by/4.0/CC BY 4.0 license.

**Figure 3 sensors-21-01051-f003:**
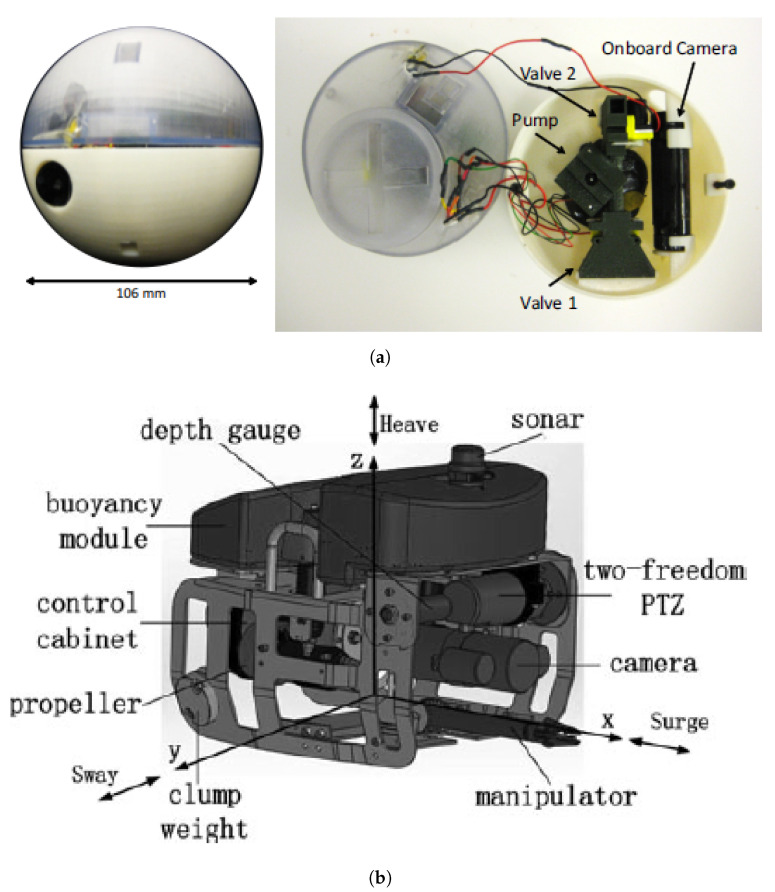
Examples of underwater robots developed for inspection of nuclear reactors and other water-filled infrastructures: (**a**) Spherical underwater robot. ©2013 IEEE. Reprinted, with permission, from Reference [60]. (**b**) Underwater robot prototype [62], reprinted by permission of the publisher (Taylor & Francis Ltd, http://www.tandfonline.com).

**Figure 4 sensors-21-01051-f004:**
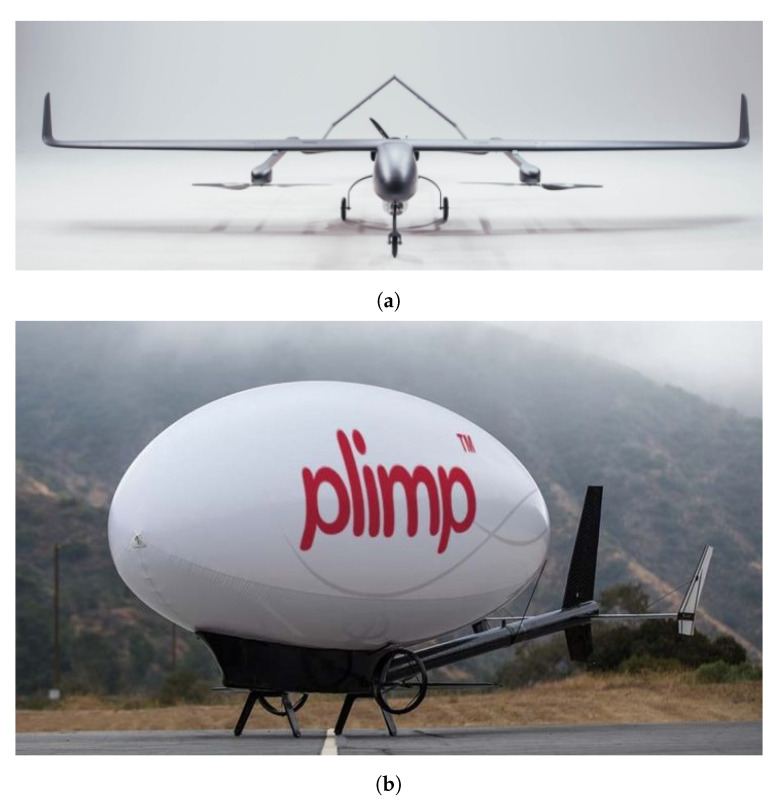
Examples of hybrid unmanned aerial vehicles (UAVs): (**a**) Hybrid vertical take-off and landing (VTOL) fixed-wing. Reproduced, with permission, from Reference [74]. (**b**) Hybrid planeblimp (PLIMP). Reproduced, with permission, from Reference [77]. Copyrighted 2021 Plimp Airships, protected by national and international patent numbers D727242, D713320, 201815761, 005665221-0001, and ZL201830546681.0.

**Figure 5 sensors-21-01051-f005:**
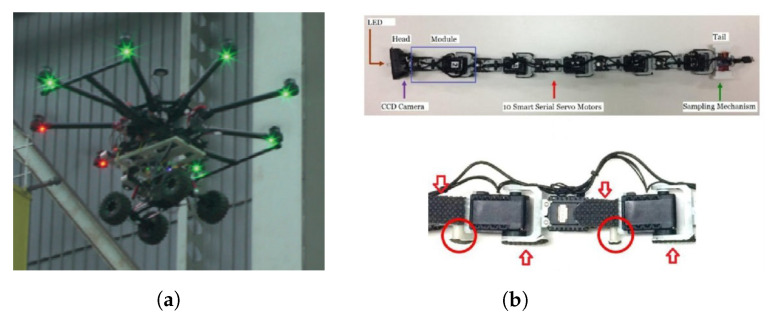
Examples of research robots: (**a**) Multi-rotor carrying a UGV. Reproduced, with permission, from Reference [80]. (**b**) Snake-like robot (**top**) and detail of the silicon pads (**bottom**). Reproduced, with permission, from Reference [84]; published by SAGE, 2018, under the terms of the https://creativecommons.org/licenses/by/4.0/CC BY 4.0 license.

**Figure 6 sensors-21-01051-f006:**
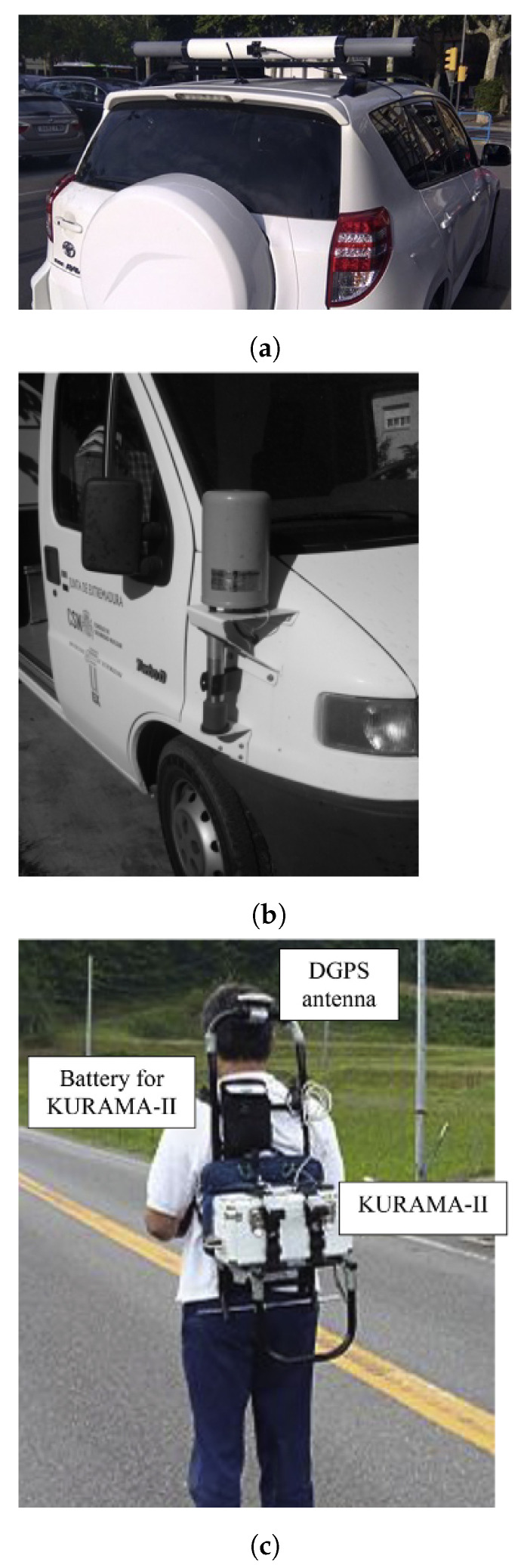
Examples of ground-based mobile detection systems: (**a**) Car-borne survey with two LaBr3 mounted in the roof. Reproduced, with permission, from Reference [124]; published by Elsevier, 2020, under the terms of the https://creativecommons.org/licenses/by/4.0/CC BY 4.0 license. (**b**) Van with a dose rate counter (top) and a NaI(Tl) spectrometer (bottom). ©2013 IEEE. Reprinted, with permission, from Reference [125]. (**c**) Backpack KURAMA-II for walk surveys using a CsI(Tl) scintillation detector. Reproduced, with permission, from Reference [128]; published by Elsevier, 2018, under the terms of the https://creativecommons.org/licenses/by-nc-nd/4.0/CC BY-NC-ND 4.0 license.

**Figure 7 sensors-21-01051-f007:**
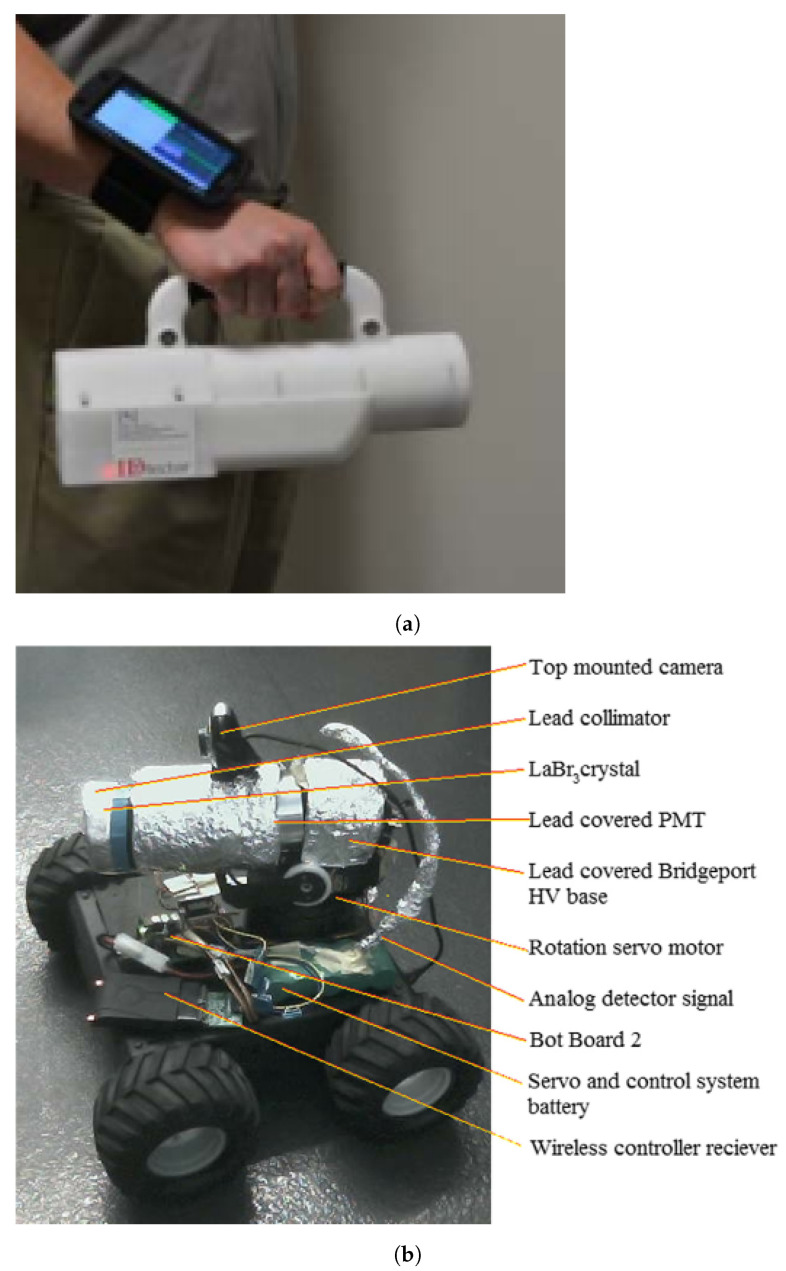
Examples of ground-based mobile detection systems: (**a**) Handheld detector using NaI(Tl) crystal (IDtector^TM^ system). ©2015 IEEE. Reprinted, with permission, from Reference [131]. (**b**) LaBr3 with a lead collimator mounted in a mobile robot. Reproduced, with permission, from Reference [134]; copyright Elsevier (2015).

**Figure 8 sensors-21-01051-f008:**
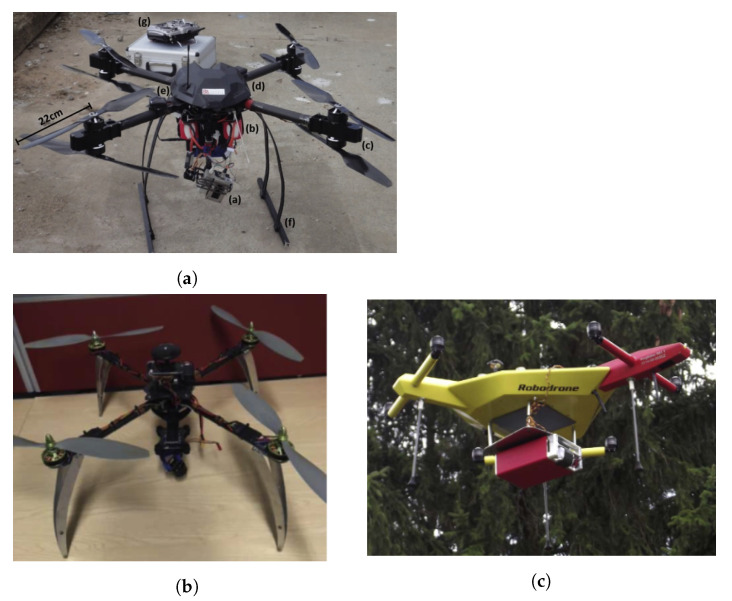
Examples of detectors coupled to air-based platforms: (**a**) Multi-rotor UAV with a Cadmium Zinc Telluride (CZT) and a single-point light detection and ranging (LiDAR) mounted in a gimbal. Reproduced, with permission, from Reference [159]; published by Elsevier, 2016, under the terms of the https://creativecommons.org/licenses/by/4.0/CC BY 4.0 license. (**b**) Multi-rotor UAV with a SIGMA-50 CsI(Tl) detector and a single-point laser rangefinder. Reproduced, with permission, from Reference [164]; published by Elsevier, 2018, under the terms of the https://creativecommons.org/licenses/by/4.0/CC BY 4.0 license. (**c**) Multi-rotor UAV with two Bismuth Germanate (BGO) detectors. Reproduced, with permission, from Reference [166]; copyright Elsevier (2018).

**Figure 9 sensors-21-01051-f009:**
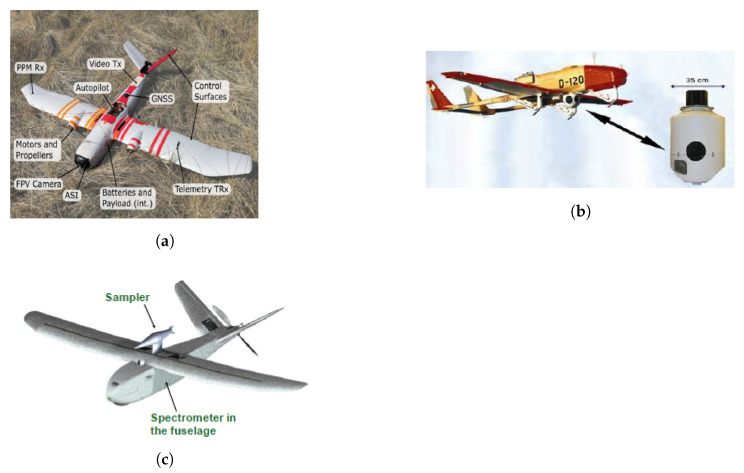
Examples of detection systems coupled to unmanned fixed-wing aircraft: (**a**) UAV with two SIGMA-50 CsI(Tl) scintillation detectors. Reproduced, with permission, from Reference [67]; published by Frontiers, 2020, under the terms of the https://creativecommons.org/licenses/by/4.0/CC BY 4.0 license. (**b**) Ranger UAV with a GM, NaI(Tl) and a CZT detection system. Reproduced, with permission, from Reference [170]. (**c**) Patria MASS Mini UAV with a CsI detector (in the fuselage) a radioactive particle air sampler (on top). Reproduced, with permission, from Reference [170].

**Figure 10 sensors-21-01051-f010:**
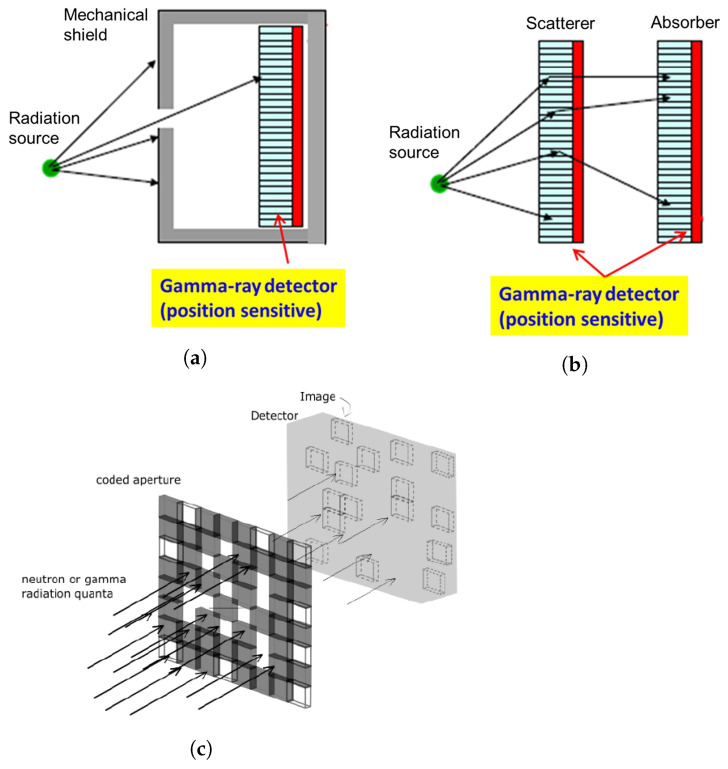
Basis of gamma cameras operation: (**a**) Pinhole camera with a single hole. Reproduced, with permission, from Reference [22]; copyright Elsevier (2013). (**b**) Compton camera. Reproduced, with permission, from Reference [22]; copyright Elsevier (2013). (**c**) Coded-aperture camera. Incident radiation (gamma-rays or neutrons) pass through a coded-aperture mask creating an image in a position sensitive detector. Reproduced, with permission, from Reference [21]; published by MDPI, 2019, under the terms of the https://creativecommons.org/licenses/by/4.0/CC BY 4.0 license.

**Figure 11 sensors-21-01051-f011:**
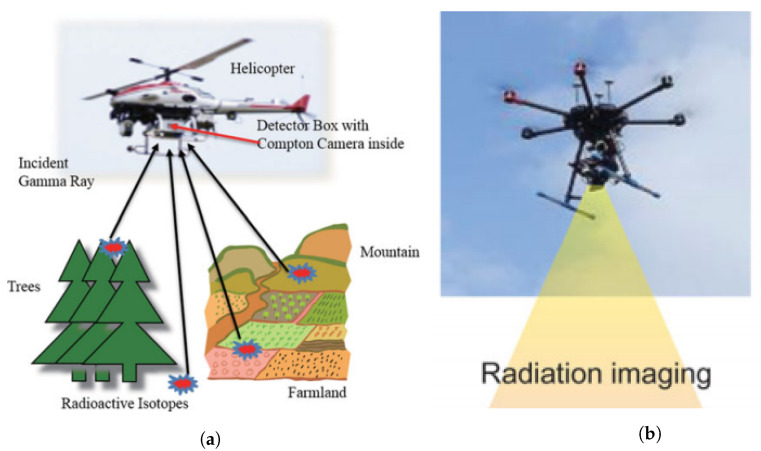
Examples of Compton cameras coupled to UAVs: (**a**) Unmanned helicopter with Compton camera composed by two arrays of Ce-doped Gd-Al-Ga-garnet (Ce:GAGG) scintillation crystals [178], reprinted by permission of the publisher (Taylor & Francis Ltd, http://www.tandfonline.com). (**b**) Multi-rotor UAV with two stages of Ce:GAGG crystals mounted in a gimbal. Reproduced, with permission, from Reference [186]; published by Taylor & Francis, 2018, under the terms of the https://creativecommons.org/licenses/by-nc-nd/4.0/CC BY-NC-ND 4.0 license.

**Figure 12 sensors-21-01051-f012:**
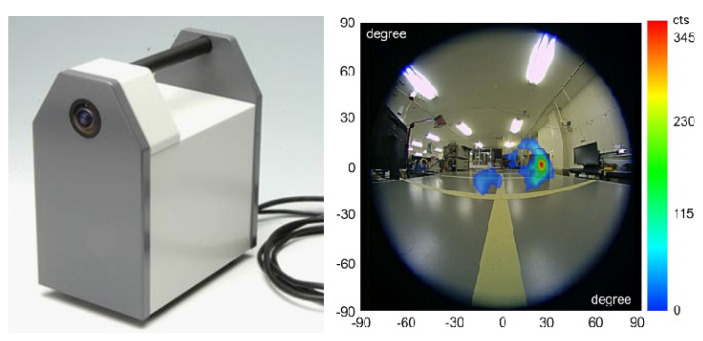
Handheld Compton camera using Ce:GAGG scintillation crystals (**left**) and reconstructed image of a ^137^Cs source (**right**). These figures were published in Reference [22]; copyright Elsevier (2013).

**Figure 13 sensors-21-01051-f013:**
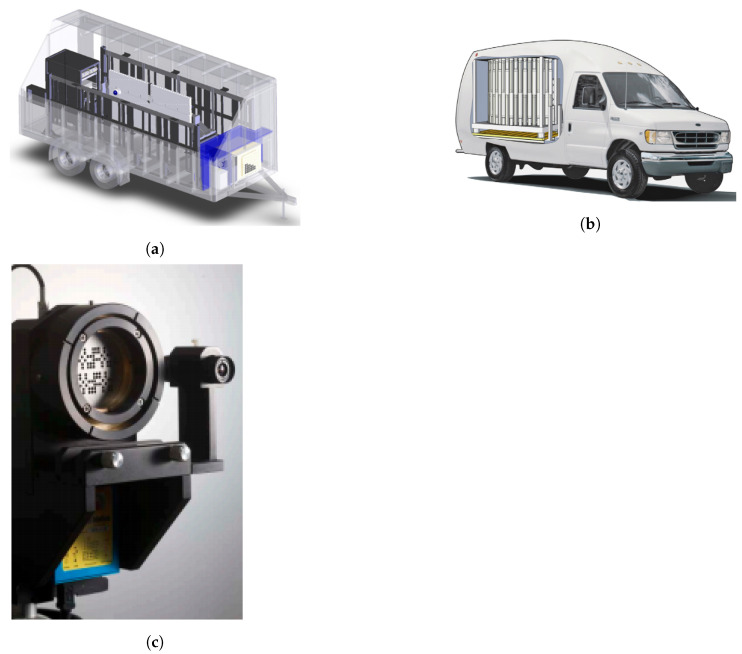
Examples of coded-aperture cameras: (**a**) Three-dimensional Stand-Off Radiation Detection System (SORDS-3D). Reproduced, with permission, from Reference [179]; copyright Elsevier (2011). (**b**) Stand-Off Radiation Imaging System (SORIS). Reproduced, with permission, from Reference [196]; copyright Elsevier (2011). (**c**) GAMPIX detector to be coupled in a multi-rotor UAV (MOBISIC project). ©2011 IEEE. Reprinted, with permission, from Reference [197].

**Figure 14 sensors-21-01051-f014:**
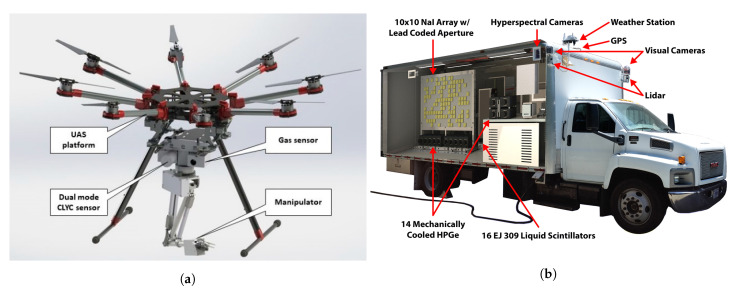
Examples of gamma and neutron detection systems: (**a**) Multi-rotor UAV with a dual-mode CLYC sensor and a manipulator. Reproduced, with permission, from Reference [201]; published by MDPI, 2019, under the terms of the https://creativecommons.org/licenses/by/4.0/CC BY 4.0 license. (**b**) Radiological multi-sensor analysis platform (RadMap) system carried by a truck showing the lead coded mask (NaI detectors behind), high-purity Ge (HPGe) detectors, and EJ-309 liquid scintillators. Reproduced, with permission, from Reference [204]; copyright Elsevier (2016).

**Figure 15 sensors-21-01051-f015:**
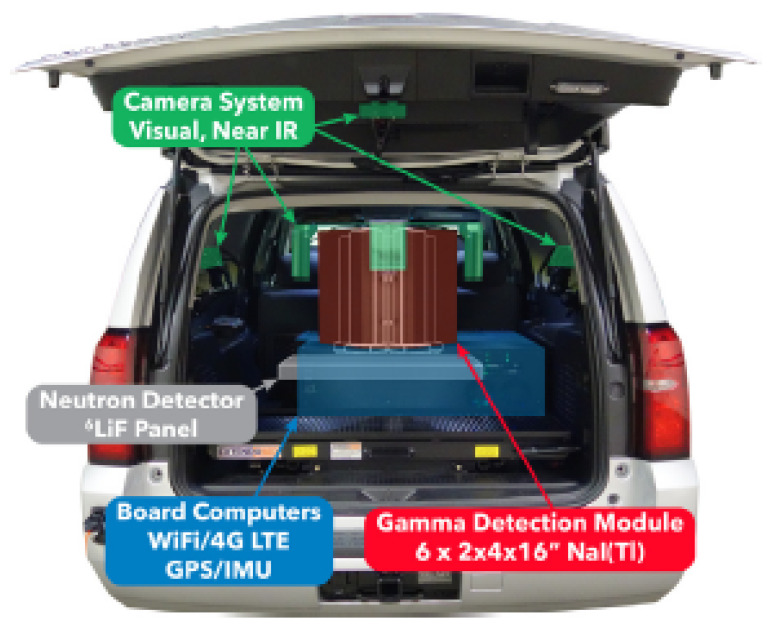
Vehicle used on the Mobile urban radiation search (MURS) project. Reproduced, with permission, from Reference [208]; copyright Elsevier (2020).

**Figure 16 sensors-21-01051-f016:**
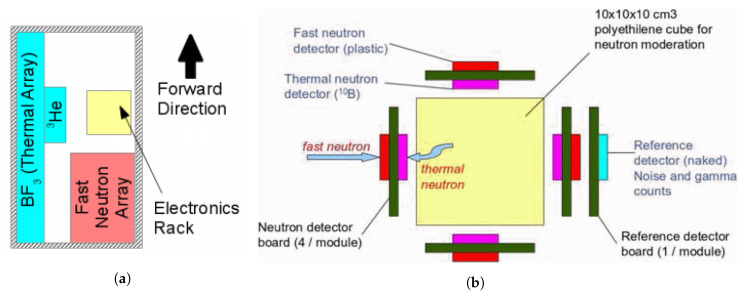
Examples of gamma and neutron detection systems: (**a**) Standoff detection of thermal and fast neutrons carried by a truck (containerized system). ©2021 IEEE. Reprinted, with permission, from Reference [209]. (**b**) Real time wide area radiation surveillance system (REWARD) project: neutron detector unit layout. ©2021 IEEE. Reprinted, with permission, from Reference [210].

**Table 1 sensors-21-01051-t001:** Standards for mobile radiation detection equipment considering scenario B. Adapted from Reference [19].

Instrument Category	IEC Standard	ANSI Standard
Alarming personal radiation devices	IEC 62401	ANSI N42.32
Spectroscopy-based alarming personal radiation detectors	IEC 62618	ANSI N42.48
Handheld instruments for the detection and identification of radionuclides	IEC 62327	ANSI N42.34
Highly sensitive handheld instruments for photon detection	IEC 62533	ANSI N42.33
Highly sensitive handheld instruments for neutron detection	IEC 62534	-
Backpack-type radiation detector	IEC 62694	ANSI N42.53
Vehicle-mounted mobile systems for the detection of illicit trafficking of radioactive materials	IEC 63121	ANSI N42.43

**Table 2 sensors-21-01051-t002:** Advantages and limitations of mobile platforms for radiation monitoring.

Platform	Advantages	Limitations	Ref.
Ground-based—Manned:			
Car, van or truck	- High spatial resolution - High payload capacity - Unfavorable weather operation - Easy to operate	- Dependent on the existent road network - Larger area coverage than foot-based - Dose risks - Radiation attenuation (vehicle structure) - Speed variable (traffic dependent)	[16,43,66]
Motorcycle ^1^	- High spatial resolution - More terrain flexibility (than cars) - Less radiation attenuation (than cars)	- Large area surveys - Payload limitation - Dose risks	
Foot-based (e.g., handheld or backpack)	- Excellent spatial resolution - Used to validate results obtained by others radiation survey methods	- Very long survey times - Large area surveys (impracticable) - Dose risks	[43,66]
Ground-based—Unmanned:			
Unmanned ground vehicle	- Medium endurance - High spatial resolution - No dose risks to operator	- Obstacles and terrain limitations (dependent on the type of UGV) - Communication problem	[10]
Air-based—Manned:			
Fixed-wing (e.g., Sky Arrow aircraft)	- Medium endurance (6 h and 1110 km) - High payload capacity - Rapid deployment (56 m/s) - Very large area coverage - “Industry standard” for large surveys	- Radiation exposure of the crew - High minimum flight altitude - High minimum flight speed - Poor spatial resolution - Needs a pilot - High operation costs	[43,66,75]
Helicopter	- Medium endurance (2 h) - Reasonably deployment time - High payload capacity (>100 kg) - VTOL and loitering - Large area coverage - Lower altitudes (compared to fixed-wing)	- Radiation exposure of the crew - Minimum flight altitude - Needs a pilot - High operation costs	[66,75,78]
Air-based—Unmanned:			
General characteristic: No dose risks to operators			
Fixed-wing (e.g., UARMS UAV)	- Medium endurance (6 h) - Reasonably deployment time (25–35 m/s) - Good fuel efficiency - Large area coverage - Large remote operation distance (100 km) - Low costs - Lower altitudes and speed than manned fixed-wing (better spatial resolution)	- Requires more training (than multi-rotor) - Payload limitation (approximately 10 kg) - Weather limitation (rain and wind) - Low costs	[17,75,78,79]
Helicopter (e.g., UHMS)	- Medium/low endurance (90 min) - VTOL and loitering - Lower altitudes and speed than unmanned fixed-wing (better spatial resol.) - High maneuverability - Low operation costs	- Requires more training (than multi-rotor) - Initial investment Compared to fixed-wing has: - Shorter remote operation distance (3–5 km) - Low payload capacity (approximately 10 kg) - Lower maximum speed (longer surveys)	[17,43,66,69,75,78]
Multi-rotor(a.k.a. drones)	- Low endurance (20 min) - VTOL and loitering - Very low altitudes and speed (high spatial resolution) - Very low costs - High maneuverability - Easy to operate	- Very short remote operation dist. (<500 m) - Low payload capacity (few kg) - Greater weather limitations	[17,43,66,69,78]
Blimp, balloon ^1^	- High endurance (Low fuel consumption) - VTOL and loitering - Low operation cost - Low vibrations, noise, and turbulence	- Poor maneuverability - Weather limitations (low wind conditions) - Low speed	[75]
Hybrid plane-blimp (e.g., PLIMP) ^1^	- Same advantages as blimps - Better maneuverability than blimps	- Weather limitations (low wind conditions) - Low speed (but faster than blimps)	[75,77]
VTOL Fixed-wing ^1^	- No runway needed - Same advantages as fixed-wing	- Complex system - Same disadvantages as fixed-wing	[71]

^1^ To the best of the authors knowledge, no work has been done using this platform for radiation monitoring.

**Table 3 sensors-21-01051-t003:** Resume of the inorganic scintillation crystal main characteristics.

Scintillator	NaI(Tl)	CsI(Tl)	LaBr3[Ce]	LaBr3[Ce+Sr]	BGO	Ce:GAGG	Ref.
Density	3.67	4.51	5.08	5.08	7.13	6.63	[43,100,101]
Effective atomic number	49.7	54	45.2		74	50.5	[102,103]
ΔE/E % at 662 keV	<7.5	6.5–8	2.6	2.2	16 ^1^	5.2	[101,104,105,106,107]
Wavelength of max emission [nm] ^2^	415	550	380	385	480	520	[100,101,106]
Photoelectron yield [% NaI(Tl)] (for γ-rays)	100	45	165	>190	20	–	[101]
Light yield (photons/keV)	38	54	63	73	8–10	46	[101,104]
Primary decay time (ns)	250	1000	16	25	300	90	[101,104]
Hygroscopic	yes	slightly	yes	yes	no	no	[101,104,106]
Self activity	no	no	yes	yes	no	no	[101,104]

^1^ Energy resolution (%) at 511 keV. ^2^ The emission wavelength (emitted light) should match the quantum efficiency of the photomultiplier (PMT) or silicon photomultiplier (SiPM).

**Table 4 sensors-21-01051-t004:** Summary of light sensors characteristics. Photosensors analysed: photomultipler (PMT), silicon photomultiplier (SiPM), avalanche photodiode (APD), and PIN diode. Adapted from Reference [111].

Light Sensor	PMT	SiPM	APD	PIN Diode
Size	Big	Small	Small	Small
Bias voltage	High	Low	Medium	Low/none
Power consumption	High	Low	High	Low
Sensitivity to microphonics	No	No	Intermediate	Yes
Magnetic field	Yes	No	No	No

**Table 5 sensors-21-01051-t005:** Selection of commercially off-the-shelf (COTS) lightweight spectrometers with potential for use in mobile platforms.

Name	Sensor Type	Sensor Size (cm^3^)	FWHM % @ 662 keV	Energy Range (keV)	Power/Signal	Weight (g)	Unit Size (mm)	Price (€)	Ref
GR-1 (Kromek)	CZT	1	<2.5	20–3000	USB (250 mW)/(USB/ MCX)	60	25 × 25 × 63	3000–9000	[18,113]
μspec (Ritec)	CZT	0.06 0.5 1.6 4	<2 <2.2 <3 <4	20–3000	USB/Micro USB	60 65 70 100	25 × 25 × 72	6500 (except 4 cm^3^ crystal)	[18,112]
MGS series (IMS)	CZT	0.06 0.5	<1.5 <2.5	30–3000	USB/USB,TTL	<50	64 × 25 × 15		[161]
SIGMA (Kromek)	CsI(Tl)	32.8 16.4	<7.2		USB (250 mW)/USB	300 200	35 × 35 × 130 35 × 35 × 105	4000–5000	[18,162]
Raspix (Crytur)	Timepix (Silicon)	14.1×14.1 mm			Ethernet (5 W) / Wi-fi or Ethernet	275	97 × 65 × 35		[163]

**Table 6 sensors-21-01051-t006:** Summary of the typical scenario characteristics and detection system performance requirements.

Scenario	A: RN Accidents and Emergencies)	B: Illicit Trafficking of SNM and Radioactive Materials	C: Nuclear, Accelerator, Targets, and Irradiation Facilities	D: Detection, Monitoring, and Identification of NORM
Typical scenario characteristics	- Actinides and other radioactive materials released - Range from low to high dose rate - Rapid deployment	- SNM and radioactive materials - Low dose rate - Rapid deployment	- Residual radioactivity (e.g., activation products in reactor materials) - Range from low to high dose rate	- U and Th series and ^40^K radionuclides - Low dose rate - Mostly slow deployment
Operational restrictions / difficulties	- Urban areas - High gamma radiation and alpha/beta contamination - Atmospheric conditions with noxious gases - High temperatures - Difficulty of access, such as in narrow spaces - Lack of visibility - Heavy loads	- Urban areas - Distinguish threat radionuclides from NORM and medical isotopes - Possible shielded or masked sources - Large standoff distances between detector and source (up to 100 m)	- High magnetic fields - High temperatures	- Measured doses near background values
Detectors desirable characteristics	- Fast response - High energy resolution - Sensitivity range from ambient background to high dose rates - Radiation hardened devices	- Meet standards referred in Section 2.2 - High energy resolution - High efficiency (low MDA)	- Fast response - Insensitive to magnetic fields Temperature stability - Radiation hardened devices	- High energy resolution - High efficiency (low MDA)
General characteristics	Reliable; robust (good mechanical and chemical properties); modular; flexibility (for different scenarios); low cost; lightweight and compact (for use in portable equipment).			

**Table 7 sensors-21-01051-t007:** Advantages and limitations of the mobile radiation detection systems described in literature used in the different scenarios (described in Section 2).

MRDS	Platforms Used according to Literature	Applications	Advantages	Limitations	Scenario	References
**Gas-filled:**					
Geiger-Muller	Manned fixed-wing, Multi-rotor, car, small UGV, PRD, smartphone	Detection and localization	Low cost and lightweight	No spectrometry information	A,C,D	[8,118,135,137,144,146,154,170]
Ionization chamber	Car	Detection and localization	-	No spectrometry information	A	[118]
Proportional counter	Van	Detection and localization	-	No spectrometry information	A	[118,125]
**Scintillators:**					
NaI(Tl)	Manned and unmanned aircrafts (fixed-wing and helicopter), van, car, backpack, handheld	Detection, identification, mapping and localization	Large volumes available commercially (allowing high efficiency detection system)	Hygroscopic	A,B,D	[38,118,119,122,123,125,130,131,150,153,154,155,170,205]
BGO	Manned fixed-wing, Multi-rotor	Detection, localization and mapping	Good sensitivity	Poor energy resolution	A,D	[154,166]
LaBr_3_[Ce]	Manned & unmanned helicopters, car, handheld, backpack, small UGV	Detection, identification, mapping and localization	Good energy resolution (high light yield), fast response, temperature stability, high magnetic field operation (with SiPM)	Intrinsic activity	A,B,C,D	[124,129,130,134,148,151,153,156,205,205]
CsI(Tl)	Unmanned fixed-wing, multi-rotor, car, van, backpack, PRD	Detection, identification, mapping and localization	Less hygroscopic and better light yield compared to NaI(Tl), lightweight particularly when coupled to SiPM	-	A,B	[66,67,111,120,121,128,164]
CsI	Unmanned fixed-wing, smartphone	Detection and mapping (monitoring)	Lightweight and low cost	Poor energy resolution	A	[139]
Ce:GAGG	Personal gamma spectrometer	Detection and identification	High sensitive and short time response (better than CsI:Tl and NaI:Tl)	-	A,B	[132]
Xe scintillator	Van	Detection and identification	Robust, vibration insensitive, non hygroscopic, better energy resolution than NaI	-	B	[207]
**Semiconductors:**					
Pin photodiode	smartphone, mobile robot, small ducted-fan UAV	Detection and mapping	Lightweight, low cost, and low power consumption	Susceptible to noise vibrations	A,C	[138,145,169]
CMOS (Timepix)	Multi-rotor	Source localization and contour mapping	Small, lightweight, low power consumption, and with a pixel matrix	-	A,B	[172]
CZT	Small UAV (multi-rotor), ground vehicles, backpack	Detection, identification, mapping and localization	High spatial resolution; Good energy resolution Reduced costs; Fast deployment	Low autonomy; Small volume sensor	A,B,D	[127,140,152,158,159,160,165,167,170,176,210,211,212,213]
CdTe	Unmanned fixed-wing, multi-rotor	Soil contamination	Good energy resolution	Only for low energy gamma-rays	A	[168,170]
HPGe	Manned fixed-wing, van, backpack	Detection and identification	Excellent energy resolution	Too heavy (cooling system)	A	[121,125,130,154]

**Table 8 sensors-21-01051-t008:** Resume of the gamma cameras described in the literature considering the scenario types (described in Section 2).

Gamma Camera	Mobile Platform	Weight	ΔE/E % (662 keV)	Acquisition Time	Scenario	Ref.
Compton Camera:
SCoTSS with arrays of CsI(Tl) and NaI(Tl) in the 3×3 configuration	Truck	15 kg	7.5–7.9	Under 3 s	A,B	[182,183]
2 arrays of Ce:GAGG coupled to SiPM	Unmanned helicopter	Lightweight	6.5	5 min	A	[178,184]
2 arrays of Ce:GAGG	Multi-rotor	1.9 kg	9	∼10 min	A	[185]
2 arrays of Ce:GAGG	Multi-rotor	1.5 kg	-	∼10 min	A	[186,187]
VCI based on HPGe	Mobile	-	-	-	A,B	[188]
VCI	Robot	-	-	-	A	[190,191]
VCI using Ce:GAGG	Crawler robot	-	-	-	A	[192,193]
2 arrays of Ce:GAGG	Handheld	1 kg	-	10 s	A	[22]
Si/CdTe Compton camera	-	10 kg	2.2	30 min	A	[194]
Polaris-H (pixelated CZT detectors)	Handheld	4 kg	1.1	∼2 min ^1^	A	[195]
Compton camera/Coded-aperture camera:
VCI (based in CZT crystals with active mask for coded-aperture mode)	Handheld	3.6 kg	2.5	-	A,B	[189]
Coded-aperture camera:
SORDS-3D (with passive mask)	Cargo trailer	-	6.13	-	B	[179]
SORIS (with active mask)	Van	-	8	-	B	[196]
GAMPIX (MOBISIC project)	Mutirotor	70 g	-	300 s	B	[197]
Pinhole camera:
Pinhole	Truck	-	-	-	A	[198]

^1^ Polaris-H may take 2 min to obtain the same localization precision as SiPM-based Compton telescope for safety and security (SCoTSS) [183].

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
