# Peer review of "State-of-the-Art Mobile Radiation Detection Systems for Different Scenarios"

_sensors, 2021, doi:10.3390/s21041051_

Round 1

Reviewer 1 Report

The review paper "State-of-the-Art Mobile Radiation Detection Systems for Different Scenarios" is well written and is suitable for publication for the special issue it has been proposed. 

It is a very extensive review paper and a layout of the paper or an index would be helpful to guide the reader also to check only selected parts, if wanted. Also, if it could be shorted a bit, without altering the content, it would be easier for the reader. 

Abstract: The abbreviations in the abstract could be removed to improve readability.

l.80: "allows" -> "may allow", depending if other shielding is present or not. 

l.103 or around there: just a note regarding the existence of similar problems in aerospace industry. Some detectors are already deployed in space, they are tailored for low weight, low power consumption and high radiation tolerance.

l.135: the 4 scenarios are mentioned again around l.165 and they could be cut here.

l.215: "Either emitters"

l.261: "Both ... and .. standards" maybe is better than "Either" here.

l.310 or around there: just for completeness Tritium, which ends up in water, is also a byproduct of accelerator based experiments.

Around page 15 or table 2: it would be nice also to know the power that the different vehicles and the different detector can provide/require to operate. 

Section 3.4: there is a variety of fast growing development around perovskite radiation detectors: the performance is not yet completely compatible to more traditional detectors, but the low costs and the large interest from the community will probably lead to good results in the next years.

l.613: also the detector efficiency is not flat in the whole energy range but varies.

l.640: The growing field of pseudo-gamma spectroscopy with plastic scintillators and have use of state of the art artificial intelligence techniques could be also mentioned here. 

l.689: Here one could expand the semiconductor section to be more balanced compared to the scintillator one. Si is the standard in semiconductor but poor efficiency for high energy gamma rays. That is why high Z materials (like CZT) are used. There are many researches made by Medipix/Medipix3 collaboration for both gamma and neutron detection. Medipix devices are also develped for space missions so they are technologically interesting too. Also now there is a new tred of trying Perosvikite semiconductor detectors (technology based on solar panel cells) for gamma ray detection. 

Section 4.2: it would have been nice to  quantify detector efficiencies for the different energy ranges (directly proportional to the dose that they are sensitive to)

Table 4: energy resolution here could be interesting 

l.731: specify "performance" at what is referred to (energy resolution, dose sensitivity?).

When I arrived to 4.3, I got a bit lost in the flow of the paper. I advise to guide the reader a bit more explaining enter how the paper is structured. It is a very long paper and maybe something could be shorted to help the reader.

It would be nice in the Conclusion section to also argument some ideas towards which the field will develop most. Remarks from the authors are somehow missing at this point or should be stressed a bit more. 

Author Response

Other changes:

For this revision it was used “track changes” and comments at Overleaf so that the Editor and Reviewers can check the changes expeditiously.

In addition to the changes made according to the Reviewers' suggestions (described in the files "Respond to Reviewer 1" and Respond to Reviewer 2 "), the following changes were also made:

Because of the lack of copyright licenses it was replaced Figure 2a: “SPOT with payload” by Figure 2a: “Anymal C from ANYbotics”, and it was removed 3 Figures: Figure 4a: “Helicopter (single main rotor)”, Figure 4b: “Multi-rotor with a LiDAR sensor”, and Figure 4c: “Honeywell T-Hawk, a ducted fan UAV”. The caption of Figure 4 also changed from “Examples of unmanned aerial vehicles (UAVs)” to “Examples of hybrid unmanned aerial vehicles (UAVs)” which includes now only 2 Figures: Figure 4a: “Hybrid VTOL fixed-wing” (figure 4d in the previous version of the article) and Figure 4b: “Hybrid plane-blimp (PLIMP)” (figure 4e in the previous version of the article).

In the end of all Figure captions it was added the copyright acknowledgement according to each copyright holder. When no instructions were given by the copyright holders it was written the following text according to MDPI journal: "Reproduced with permission from [author], [book/journal title]; published by [publisher], [year].” at the end of the Figure caption.

Changed the bibliographic reference of Figures 1a and 1b to the original source (reference was also added to bibliography):

Kim, S.; Jung, S.H.; Lee, S.U.; Kim, C.H.; Shin, H.C.; Seo, Y.C.; Lee, N.H.; Jung, K.M. Application of Robotics for the Nuclear Power Plants in Korea. In Proceedings of the 2010 1st International Conference on Applied Robotics for the Power Industry (CARPI 2010); IEEE: Montreal, QC, Canada, October 2010; pp. 1–5. doi: 10.1109/CARPI.2010.5624417

The figure´s quotes of 10b and 10c were exchanged between them.

Reviewer 2 Report

This is a fundamental overview work presenting the current developments in the mobile radiation detection (MRD) technology. It overviews 223 bibliographic references, with proper analysis and discussion. I recommend publication of this work after minor revision by taking into account my seven comments listed below.

Comments:

  1. Page 2, line 31: Please correct the date of the World Trade Center attacks. The correct date is September 11, 2001. Unfortunately, in your text we read ”the attack to the World Trade Center towers (September 9, 2001).”
  2. Line 43: Missing a point sign in “…radioactive material They are located…” Please insert it, changing to “…radioactive material. They are located…”
  3. Line 84: Proposed to change “indicated” to “preferable” in the phrase: “gamma-rays and neutrons are specially indicated for mobile detection systems.”
  4. Lines 169 and 170: In the following clause the double use of verb is found: “During the Fukushima nuclear accident it was released radioactive isotopes (mainly 137Cs, 134Cs and 131I) were released to the atmosphere…” Please delete duplication, making it like this: “During the Fukushima nuclear accident, radioactive isotopes (mainly 137Cs, 134Cs and 131I) were released to the atmosphere…”
  5. For clearer structuring the text and further navigation in the paper, it is highly recommended to specify the lettered sequence of the scenario types defined in lines 162-165 in the subtitles 2.1, 2.2, 2.3, 2.4. For example, subtitle 2.1 could be read “Radiological and Nuclear Accidents and Emergencies – Scenario A”, and so on, for 2.2 add Scenario B, for 2.3 – scenario C, and finally, for 2.4 - D. Mentioning the scenario letters on the subtitles could be useful for selection which MRD device should be used for the particular scenario of radiation detection. Scenario depends on the radiation environment and detecting subjects.
  6. Lines 292-293: Please remove redundant verb in the sentence: “In [33] it is presented some R&D needed in the decommissioning of nuclear facilities is discussed.” Therefore, the sentence with corrected English could be like this: “In [33] some R&D activities needed for the decommissioning of nuclear facilities are discussed.”
  7. Lines 311-312: Please refer as well to the reference presenting the decommissioning process of fusion machines, e.g. “Decommissioning of the Tokamak Fusion Test Reactor” DOI:1109/FUSION.2003.1426633

Author Response

(The authors gave the same response as above.)
